# Living Cells and Cell-Derived Vesicles: A Trojan Horse Technique for Brain Delivery

**DOI:** 10.3390/pharmaceutics15041257

**Published:** 2023-04-17

**Authors:** Ante Ou, Yuewei Wang, Jiaxin Zhang, Yongzhuo Huang

**Affiliations:** 1State Key Laboratory of Drug Research, Shanghai Institute of Materia Medica, Chinese Academy of Sciences, Shanghai 201203, China; 2University of Chinese Academy of Sciences, Beijing 100049, China; 3Zhongshan Institute for Drug Discovery, Shanghai Institute of Materia Medica, Chinese Academy of Sciences, Zhongshan 528437, China; 4NMPA Key Laboratory for Quality Research and Evaluation of Pharmaceutical Excipients, Shanghai 201203, China; 5School of Pharmaceutical Sciences, Southern Medical University, Guangzhou 510515, China

**Keywords:** brain drug delivery, blood–brain barrier (BBB), Trojan horse delivery systems, cells, extracellular vesicles, cell membranes

## Abstract

Brain diseases remain a significant global healthcare burden. Conventional pharmacological therapy for brain diseases encounters huge challenges because of the blood–brain barrier (BBB) limiting the delivery of therapeutics into the brain parenchyma. To address this issue, researchers have explored various types of drug delivery systems. Cells and cell derivatives have attracted increasing interest as “Trojan horse” delivery systems for brain diseases, owing to their superior biocompatibility, low immunogenicity, and BBB penetration properties. This review provided an overview of recent advancements in cell- and cell-derivative-based delivery systems for the diagnosis and treatment of brain diseases. Additionally, it discussed the challenges and potential solutions for clinical translation.

## 1. Introduction

Brain diseases, including neurodegenerative disorders, cerebrovascular diseases, and brain cancers, remain a leading cause of disability and mortality worldwide [1]. However, the development of new therapeutics for brain diseases is challenging, because the blood–brain barrier (BBB), a highly selective semipermeable interface between systemic blood circulation and the brain parenchyma, impedes the transportation of drug molecules across it [2,3].

The BBB is mainly comprised of tightly connected brain capillary endothelial cells functionally supported by pericytes and astrocytic perivascular endfeet [4]. The BBB represents a physical and biochemical barrier that tightly controls the transport of ions, molecules, and cells between the blood and the brain to maintain intracranial homeostasis [5], and meanwhile, the BBB protects the brain from the attack of circulating toxins and pathogens. However, this highly selective physiology barrier excludes almost all macromolecular therapeutics and over 98% of small-molecule drugs from entering the brain [6]. Because of restricted transportation across the BBB, FDA-approved drugs for central nervous system diseases (CNS) are generally limited to small lipophilic molecules [7]. Although nanoparticles have been explored as a non-invasive mechanism for BBB drug delivery, no nanoparticulate formulations for CNS diseases have been approved by the FDA because of concerns about targeting ability, toxicity, reproducibility, and large-scale production [7,8].

Recently, cell-based delivery systems manufactured from biological materials such as cells, cell components, or cell-derived vesicles have received increasing interest because of their bio-mimic properties [9]. With intrinsic abilities to penetrate the bio-barriers (e.g., the BBB) and excellent characteristics to target the lesioned brain areas, cell-based delivery systems can act as “Trojan horses” to deliver drugs into the brain [9], while maintaining their original biological functions and properties after drug-loading processes [10,11]. Compared with conventional drug delivery systems, these delivery platforms have advantages over conventional drug systems, such as long circulation time, maximal biocompatibility, and minimal toxicity [10,11,12]. In particular, these “Trojan horses” not only serve as drug carriers to mediate BBB delivery but also act as synergistic therapeutics to ameliorate certain brain diseases, owing to their intrinsic properties and inherent active components [13,14].

In this review, we provide an overview of the prospects of cell- and cell-derivative-based delivery systems for overcoming the BBB and the research progress on the diagnosis and treatment of brain diseases (Figure 1). Finally, the existing challenges and potential solutions for the clinical translation of these Trojan delivery systems are discussed.

## 2. Cell-Mediated BBB-Crossing Delivery

Cell-based delivery systems have the advantages of good biocompatibility, low immunogenicity, specific tissue-homing characteristics, and inherent ability to penetrate various biological barriers (e.g., the BBB) [10]. For example, stem cells (e.g., mesenchymal stem cells, neural stem cells, and adipose-derived stem cells), mononuclear phagocytes (e.g., monocytes, macrophages, and microglia), neutrophils, and T cells have been widely used to prepare cell-based delivery systems for brain targeting (Table 1).

The candidate cells as carriers should possess good drug-loading potential. The drug-loading process can be based on the physicochemical nature of the target drugs and the biological properties of the candidate cells. General ways for drug loading into carrier cells include cell surface coupling via covalent or non-covalent methods; internalization via endocytosis, phagocytosis, or macropinocytosis; direct cellular entrance through direct translocation, lipid fusion, electroporation, or microinjection; and antibody-antigen recognition [15,16].

Internalization of drugs into the carrier cells is the most used loading method. For phagocytes (e.g., macrophages and neutrophils), their phagocytic properties can be a convenience for drug loading [17]. For cells of low phagocytic ability (e.g., red blood cells and T lymphocytes), electroporation is an alternative method for loading macromolecular drugs [18], but it may cause irreversible cellular damage [19]. Notably, functional molecules (e.g., cell-penetrating peptides, targeting ligands, or “eating-me” signal molecules) can mediate drug loading into cells and these methods have been under extensive investigation [20,21,22]. In addition, electrostatic interaction between positively charged surfaces of nanoplatforms and negatively charged plasma membranes of the candidate carrier cells could also increase the loading capacity [23].

The mechanisms involved in the carrier cell migration across the BBB include rolling, activation, arrest, crawling, and diapedesis [24,25]. After reaching around the lesioned brain areas, cell-based drug delivery systems release therapeutic cargos to exert therapeutic effects. Various cell-based brain-targeting delivery systems are presented and discussed in the following chapter.

### 2.1. Stem Cells

#### 2.1.1. Mesenchymal Stem Cells (MSCs)

MSCs are pluripotent stem cells with tumor-homing capacity that can transmigrate the BBB through paracellular and transcellular routes through G-protein coupled receptor and integrin very late antigen-4 and its ligand vascular cell adhesion molecule-1 (VLA-4/VCAM-1) dependent mechanisms [26,27]. MSCs engineered with multifunctional mesoporous silica nanoparticles incorporating FITC/NIR dye ZW800/Gd^3+^/^64^Cu were conducted for glioma delivery and multimodal optical/MR/PET imaging (Figure 2) [28]. The mesoporous silica nanoparticles could be rapidly internalized by MSCs via phagocytosis, pinocytosis, and especially receptor-mediated endocytosis without impairing the proliferation and tumor tropism of MSCs (Figure 2B,C) [28]. The resulting nanoparticle-loaded MSCs achieved 5.2-fold higher accumulation in the glioblastoma than the free nanoparticles (Figure 2D), exemplifying an MSC-based platform characterized by combining the superiority of MSCs and nanoparticles for glioblastoma targeted delivery [28].

#### 2.1.2. Neural Stem Cells (NSCs)

NSCs are multipotent cells originating in the central nervous system with the potential to generate neurons and glial cells. NSCs exhibited targeted migration toward glioblastoma after being injected intravenously [29]. The engineered NSCs were prepared by transduction with adenovirus for expressing a rabbit carboxylesterase or a modified human carboxylesterase to convert prodrug CPT-11 (irinotecan) to its active metabolite SN-38 in situ for glioma treatment [30]. The glioma-bearing mice that received the carboxylesterase-expressing NSCs in combination with CPT-11 had significantly higher SN-38 concentrations in the glioma than mice that only received CPT-11, but not in the normal brains [30].

#### 2.1.3. Adipose-Derived Stem Cells (ADSCs)

ADSCs are mesenchymal stem cells obtained from adipose tissue with the capacity of homing to brain cancer via stromal cell-derived factor-1α/C-X-C chemokine receptor type 4 (SDF-1α/CXCR4) signaling axis, thus could serve as therapeutic vehicles for targeted brain tumor delivery [31]. The engineered ADSCs carrying nanotherapeutic payloads that co-loaded oleic acid-coated superparamagnetic iron oxide nanoparticles (SPIONs) and paclitaxel were prepared for thermal/chemotherapy of glioblastoma [32]. This ADSC-based delivery strategy could improve the accumulation of therapeutic nanoparticles in brain tumors and achieve pronounced therapeutic efficacy against orthotopic glioblastoma [32].

### 2.2. Immune Cells

The inflammatory responses occurring in CNS diseases (e.g., ischemic stroke [20,21] and brain cancer [17,33]) usually involve the recruitment of leukocytes, which possess the native ability to penetrate through the BBB and the blood–brain tumor barrier (BBTB), thereby being explored as “Trojan horses” for brain drug delivery.

#### 2.2.1. Neutrophils (NEs)

Neutrophils are the most abundant innate immune cells (making up 50–70% of circulating leukocytes) and the first type of leukocyte recruited to the lesions of inflammation and tumorigenesis, making them prospective in cell-based drug delivery. With the ability to traverse the BBB/BBTB [34,35], NEs have well been explored as drug carriers to target brain lesions.

Neutrophils internalizing doxorubicin-loaded magnetic mesoporous silica nanoparticles (ND-MMSNs) have been reported for the therapy of postsurgical glioma (Figure 3) [33]. ND-MMSNs could retain the native functions of neutrophils without compromising the host cell’s viability, chemotactic capability, and BBB/BBTB penetrating ability [33]. The glioma resection surgery increased the levels of proinflammatory cytokines (e.g., TNF-α, IL-6) in the brain, which promoted the recruitment of systemic administrated ND-MMSNs to the residual glioma site (Figure 3B–D), improved survival rate, and delayed glioma relapse in the surgically treated glioma mice (Figure 3E) [33]. Similar mechanisms were also revealed in the application of the NEs carrying paclitaxel-loaded cationic liposomes (PTX-CL/NEs) for the suppression of postoperative malignant glioma recurrence [17]. Notably, PTX-CL/NEs presented no significantly enhanced accumulation in the brain of the untreated glioma-bearing mice, and little suppression of gliomas growth, suggesting that amplification of the inflammatory signals after surgery was the major driving force for the NE-based targeted delivery [17]. Although chemotaxis of these NE-based drug carriers to the inflamed brain tumor led to delaying tumor relapse after surgery, the mice could not be fully cured [17]. Therefore, there is much room for enhancing drug loading and navigation capacities. Bacteria membrane camouflaging could enhance the phagocytosis by neutrophils, thus enhancing drug loading capacity [36]. The neutrophils phagocytizing Fe_3_O_4_ magnetic nanogels showed the navigating capability upon exposure to a rotating magnetic field, with increased accumulation in the diseased regions, as well as facilitated by the chemotaxis-driving BBB penetration of neutrophils for targeted drug delivery [36].

In addition, neutrophil-based vehicles have been explored for glioblastoma multiforme photoacoustic imaging [37], and cerebral ischemia reperfusion injury treatment by enhancing the accumulation of puerarin-loading liposomes in the brain parenchyma [23].

However, the use of cell-based vehicles is limited by invasive procedures to harvest carrier cells, complicated fabrication processes, and unsuitability for chronic diseases [38]. To avoid these disadvantages, endogenous circulatory leukocytes have been applied for serving as therapeutics carriers, which is achieved by modifying the nanocarrier surface with the targeted moieties (e.g., PGP [20] and RGD peptides [21]) orchestrated to bind with some specific cell surface receptors. After intravenous administration, the PGP-modified and catalase-encapsulating nanoparticles (*cl* PGP-PEG-DGL/CAT-Aco NPs) were internalized by the circulating neutrophils via endocytosis and micropinocytosis and transported across the BBB to the ischemic subregion; the delivery method could enhance the therapeutic efficiency of neuroprotectant in the middle cerebral artery occlusion (MCAO) mice and extend the therapeutic time window in acute cerebral ischemia [20]. Platelet membrane-camouflaged nanoparticles have been reported to deliver miRNA-Let-7c for the treatment of cerebral ischemic injury; the platelet membrane-cloaked nanoparticles were endocytosed by the circulating neutrophils and delivered to the area of the ischemic injury [39].

#### 2.2.2. Mononuclear Phagocytes

Monocytes are the largest leukocytes circulating in the bloodstream that originate in the bone marrow and can differentiate into different types of monocyte subsets (e.g., macrophages, and dendritic cells (DCs)) with various immunological functions. Monocytes can be recruited to the glioma by tumor-associated cytokines and chemokines [40]. The infiltration of peripheral monocytes/brain-intrinsic microglia to the glioma sites is mediated by the interaction between C-C chemokine receptor type 2 (CCR2) expressed on monocytes and C-C motif chemokine ligand 2 (CCL2/MCP-1) expressed by glioma cells [41].

It is an intriguing strategy to use monocytes as Trojan vehicles to sneak chemotherapy drugs into brain tumors. Of note, it is well-documented that monocytes and macrophages could well tolerate loaded chemotherapy drugs enveloped in or attached to nanoplatforms [22,42]. The monocytes/macrophages internalizing nano-doxorubicin maintained good viability and tumor tropism because intracellular particles were mainly sequestered in the lysosomal compartment, and kept away from the nucleus where doxorubicin exerts its toxic action; lysosomal sequestration may further combine with the induction of self-protective autophagy to underly the bio-tolerance of the carrier cells to nano-doxorubicin [13,42]. The doxorubicin-loaded poly(lactic-co-glycolic acid) (PLGA) nanoparticles efficiently killed the cancer cells but exhibited reduced cytotoxicity on the carrier macrophages [43].

Circulating monocytes serve as a primary source of tumor-associated macrophages (TAMs) in glioblastoma [41,44], and thus represent a promising cell carrier candidate for brain tumor-targeted delivery. Monocytes were employed to deliver the conjugated polymer nanoparticles (CPNs) to the glioblastoma spheroids and brain tumors [45]. In addition, lipopolysaccharide (LPS)-activated monocytes with preferential differentiation toward M1 macrophages showed an enhanced ability for CPN phagocytosis [45]. Nano-doxorubicin-loaded monocytes were capable of infiltrating the 3D glioblastoma spheroids and orthotopic glioblastoma xenografts and releasing cargo drugs therein, which subsequently killed the glioblastoma cells [42]. However, the clinical translation of monocyte-based drug delivery strategies may be hindered by the inconvenience involved in monocyte extraction and preparation. It would be more clinically practical to have nano-based drugs administrated in the blood circulation and get loaded by the circulating monocytes [42].

Bone marrow-derived macrophages have also well been explored as cellular carriers for the delivery of nanoparticles. As professional phagocytes of the innate immune system, macrophages could rapidly accumulate in the inflamed pathological sites [46,47]. The macrophage-camouflaged afterglow nanocomplex (UCANPs@RAW) was prepared for brain inflammation visualization by taking advantage of the immune-homing capacity of RAW (Figure 4) [48]. Transgenic macrophages expressing neurotrophic factors could infiltrate the degenerating CNS regions and ameliorate neurodegeneration in Parkinson’s disease [49].

Tumor progression commonly includes a massive accumulation of macrophages in the tumor microenvironment, which constitute up to 40% of tumor mass in the case of glioblastoma [50]. The Fe_3_O_4_-Cy5.5-loaded macrophages were able to target glioma and achieve multimodal imaging and effective photothermal therapy [51]. Microglia, brain-resident macrophages, could mediate brain delivery of the paclitaxel-encapsulating liposomes for glioma therapy [22]. The loaded drugs were transferred from the microglia carrier cells to the glioma cells via extracellular vesicles (EVs) and tunneling nanotubes (TNTs) [22]. The engineering microglia activated by citric-acid-coated iron oxide nanoparticles could accumulate in and around glioma tissue for intraoperative imaging [52].

**Table 1 pharmaceutics-15-01257-t001:** Cell-mediated BBB-crossing delivery.

Cell Types	Disease Model	Cargo	Loading Mechanism	Administration Method	Release Mechanism	In Vitro/In Vivo Improvement	Refs.
MSC	Orthotopic glioblastoma model	MSNs incorporating FITC/NIR dye ZW800/Gd^3+^/^64^Cu	Phagocytosis, pinocytosis, and receptor-mediated endocytosis	IV	NA	Achieved 5.2-fold higher glioblastoma accumulation than the free nanoparticles	[28]
Human NSC line (HB1.F3.CD)	U251 glioma	Adenoviral vectors encoding rCE or hCE1m6	Adenovirus transfection	IV or intracranial injection	NA	Allow tumor-localized conversion of irinotecan to its active metabolite and increase the therapeutic efficacy	[30]
ADSC	ALTS1C1 glioma	SPION/PTX-loaded NPs	Endocytosis	IV	HFMF-mediated intercellular drug transport	Exhibited a 4-fold higher therapeutic index on glioma-bearing mice compared to the typical chemotherapy using temozolomide	[32]
Neutrophil	Incomplete resection glioma model	D-MMSNs	Phagocytosis	IV	NET formation	Increased accumulation in the brain tumor; improve survival rate and delay glioma relapse	[33]
Neutrophil	Glioma surgical resection model	PTX-CL	Endocytosis	IV	NET formation	Slowed the glioma recurrence, and improved survival rates	[17]
Neutrophil	Glioma surgical resection model	PTX-loaded EM@nanogels	Phagocytosis	IV	NA	Improved the accumulation of PTX at the glioma sites, and extended the median survival time	[36]
Neutrophil	Glioblastoma multiforme model	A molecular photoacoustic imaging probe TFML	Lipid insertion into the neutrophil membrane	IV	NA	Enhanced the photoacoustic signals for glioblastoma multiforme detection	[37]
Neutrophil	Focal cerebral ischemia-reperfusion	Cationic liposomes of puerarin	“Proton sponge effect” exhibited by cationic liposomes	IV	NET formation	Enhancing the therapeutic potential of puerarin to promote neuroprotection	[23]
Neutrophil	Cerebral ischemic	PM-camouflaged nanoparticles containing miRNA-Let-7c	Binding of P-selectin on the PM with PSGL-1 on the surface of neutrophils	IV	pH-responsive release	Achieved brain delivery of miRNA-Let-7c for the targeted regulation of neurons and microglia	[39]
Neutrophil	Cerebral ischemia	cl PGP-PEG-DGL/CAT-Aco NPs	NPs were internalized by circulating neutrophils via endocytosis and micropinocytosis	IV	Transient intercellular connections and exosomes	Enhanced the delivery of NPs across the BBB in vitro and in vivo, and improved the therapeutic outcome of cerebral ischemia	[20]
Monocyte/neutrophil	Cerebral ischemia	ER-cRGDLs	Internalization triggered by the binding of cRGD with integrin αvβ1 on the surface of monocyte/neutrophil	IV	NA	Facilitated the delivery of drugs across the BBB in vitro and in vivo, improved the neuroprotection effect of ER in late-stage of ischemia, and extended the therapeutic window	[21]
Monocyte	Glioblastoma	CPNs	Phagocytosis	IV	Exosome-mediated extracellular release	Enhanced the delivery of CPNs into glioblastoma spheroids and the orthotopic model, and improved the photodynamic therapy in glioblastoma spheroids	[45]
Monocyte	Glioblastoma	Nano-doxorubicin	NA	IV	Lysosomal exocytosis	Improve spheroids infiltration and brain tumor drug delivery	[42]
RAW	Acute nerve inflammation induced by LPS	UCANPs	Phagocytosis	IV	NA	UCANPs@RAW could penetrate the BBB and image the deep inflamed region in the brain	[48]
HSC transplantation-based macrophage	Parkinson’s Disease	Lentiviral vectors expressing GDNF	Lentiviral transfection	IV	NA	GDNF-expressing macrophages infiltrated degenerating PD-like brains in mouse models, increased GDNF levels in the midbrain, and improved motor and non-motor dysfunction via the neuroprotective effects of GNDF	[49]
Macrophage	C6 glioma model	Fe_3_O_4_-Cy5.5	Phagocytosis	IV	NA	Achieved deep glioma accumulation for multimodal diagnosis, imaging-guided surgery, and photothermal therapy	[51]
BV2 microglial cell	Orthotopic glioblastoma mouse model	CIONPs and DiD	NA	Internal carotid artery injection; IV	NA	DiDBV2-Fe efficiently accumulated in the brain tumor for fluorescence-guided resections	[52]
BV2 microglial cell	U87 and GL261 glioma model	Liposomes encapsulating paclitaxel	Phagocytosis	IV	Extracellular vesicles and tunneling nanotubes	Achieved brain-targeted delivery, and suppressed tumor progression	[22]
M1 macrophage	U87 glioma model	DOX-loaded PLGA nanoparticles	Phagocytosis	IV	Exocytosis in exosome form	Improve brain tumor distribution and enhance the anti-glioma effect	[13]

MSC, mesenchymal stem cells; MSNs, mesoporous silica nanoparticles; IV, intravenous injection; rCE, rabbit carboxylesterases; hCE1m6, modified human carboxylesterases; ADSC, adipose-derived stem cells; NPs, nanoparticles; SPION/PTX-loaded NPs, nanoparticles co-loading paclitaxel and oleic acid-coated superparamagnetic iron oxide nanoparticles; HFMF, high frequency magnetic field; D-MMSNs, doxorubicin-loaded magnetic mesoporous silica nanoparticles; NET, neutrophil extracellular trap; PTX-CL, paclitaxel encapsulated cationic liposomes; EM, *E. coli* membrane vesicles; PM, platelet membrane; PSGL-1, P-selectin glycoprotein ligand 1; cl, cross-linked; PGP peptide, Ac-Pro-Gly-Pro-His-His-His-Lys-Cys; DGL, dendrigraft poly-L-lysine; CAT, catalase; Aco, cis-aconitic anhydride; ER, edaravone; cRGD, cyclo (Arg-Gly-Asp-D-Tyr-Lys); cRGDLs, liposomes coupled with cRGD; CPNs, conjugated polymer nanoparticles; UCANPs, a photochemical afterglow nanocomplex; HSC, hematopoietic stem cell; GNDF, glial cell-line-derived neurotrophic factor; CIONPs, citric-acid-coated iron oxide nanoparticles; DiDBV2-Fe, iron-oxide nanoparticle engineered BV2 microglial loading near-infrared fluorescent dye DiD; DOX, doxorubicin; PLGA, poly(lactic-co-glycolic acid).

Macrophages can be divided into non-activated (M0), classically activated (M1), and alternatively activated (M2) macrophages based on phenotype and function [53]. Non-stimulated M0 macrophages are less phagocytic; M2-like tumor-associated macrophages are known to cause tumor immunosuppression and thereby promote tumor progression; whereas the pro-inflammatory M1 macrophages mediate the elimination of pathogens and cancer cells with a potent phagocytic capacity to internalize particles and are suitable for serving as cell carriers [13]. M1 macrophage was reported to boost the internalization of the doxorubicin-loaded PLGA nanoparticles and carry them to penetrate the BBB and accumulate in the brain tumor [13].

#### 2.2.3. T Cells

T cells are a class of attractive biological carriers for their intrinsic ability to infiltrate across the BBB and migrate into inflamed lesions [54] and invasive brain tumors [55]. With the potent ability to penetrate the BBB, T cells can also reach the non-inflamed area, which may be used as carriers for diagnostic purposes [54,56].

Myelin-specific T cells could load and deliver the magnetite nanoparticles into the CNS in both naive mice and experimental multiple sclerosis mice, which was identified by immunohistological methods [57]. The magnetite core hybridized with polymeric compounds was feasible for conjugation with drugs and could be internalized by low-phagocytic T cells, thus representing a chance of utilizing T cells as a brain-targeting delivery system [57]. CD4^+^ helper T cells were used to transport the covalently surface-modified polymer nanoparticles into the brain parenchyma of the mice after systemic administration via the carotid artery [58]. In addition, by using an in vitro genetic reprogramming method, T cells were electroporated to load granulocyte-macrophage colony-stimulating factor (GM-CSF) RNA while retaining their inherent effector functions and achieving effective delivery of GM-CSF to the brain tumors [18]. Intravenous injection of the GM-CSF-expressing T cells significantly prolonged overall survival in a murine brain tumor model and the mechanisms were associated with the increased levels of interferon-gamma (IFN-γ) in the tumor microenvironment [18].

## 3. EV-Mediated BBB-Crossing Delivery

Extracellular vesicles (EVs), including exosomes, ectosomes (microvesicles, or microparticles), and apoptotic bodies, are membrane-coated nanoparticles released from various types of cells into the extracellular space, which play a key role in cell-to-cell communication through molecular transport and information transfer [59]. Given their cargo-carrying nature, good biocompatibility, low immunogenicity, biodegradability, and specific homing ability, EVs have been considered promising drug delivery vehicles. Their ability to penetrate the BBB can be applied for targeted delivery to treat brain diseases, such as neurodegenerative diseases, cerebrovascular disease, and brain tumors [14,60,61]. The BBB penetration mechanisms of EVs mainly involve (1) receptor-mediated endocytosis, (2) macropinocytosis, and (3) lipid raft-mediated endocytosis [11].

The preparation process of EV-based delivery systems for targeting brain diseases includes four main procedures: selection of parent cells, isolation of EVs, surface functionalization, and drug loading [59]. EVs may have natural organ tropism properties as their parent cells. The isolation of EVs is often achieved by ultracentrifugation and density gradient ultracentrifugation [59]. To improve the BBB permeation ability of the EVs, surface modification is often applied. Based on the lipid bilayer structure of EVs, hydrophilic cargoes can be incorporated into the inner compartment of EVs and hydrophobic cargoes can be packaged into the lipid bilayers [11]. Bioactive molecules can be loaded into EVs via several methods, including endogenous and exogenous drug loading. For the endogenous drug-loading approaches, EVs containing the desired molecules from parent cells are produced via the biogenesis process [11,59]. The most common strategy is to transfect the parent cells with a specific gene encoding nucleic acid sequence or a targeted protein [60,62,63]. This method is facile and feasible for producing the targeted nucleic acids and proteins but the contamination of transfection reagents might be a concern [11,59]. The exogenous approaches include passive diffusion into the EVs via co-incubation, physical binding or chemical conjugation of drugs to the EV surfaces, and transiently opening the EV membrane to encapsulate drugs into the vesicles by physical (e.g., extrusion, sonication, electroporation, and freeze-thaw cycles) or chemical strategies (e.g., saponin treatment) [11,59,64]. The application of exogenous methods depends on the properties of drugs and the source of EVs.

The targeting effects of EVs, to a large extent, depend on their surface molecules inherited from the parent cells [64]. Thus the origin of EVs is a pivotal factor affecting their targeting ability. In the next section, we briefly review the application of EVs from different types of cells, plants, and other sources for brain delivery (Table 2).

### 3.1. Stem Cells

Stem cell-based therapy can benefit from the self-renewal ability and differentiation potential [65]. However, its clinical application is limited by the uncontrollable differentiation process and the potential risk of immune rejection and mutagenicity [66]. EVs secreted by stem cells have been found to have similar effects to the sourced cells [67]. Therefore, stem cell-derived EVs, as cell-free agents with lower cost and higher biosafety than stem cell-based therapy, may achieve a variety of therapeutic effects on brain diseases [68].

#### 3.1.1. Mesenchymal Stem Cells (MSCs)

MSCs possess immunomodulatory properties and high regenerative capacity, which are related to their paracrine activity and EV secretion [67,69,70]. EVs contain soluble factors and metabolites secreted by MSCs, and hence they can regulate multiple physiological processes, such as cell proliferation, differentiation, migration, and immunomodulation [11]. MSC-EVs may inherit the ability to penetrate the BBB and target the diseased brain areas of the parent MSCs, and can serve as transport vectors for brain delivery [67].

In an experimental Alzheimer’s disease treatment, mesenchymal stem cell-derived exosomes (MSC-Exos) were utilized to decrease the Aβs plaque burden and rescue the neuronal memory/synaptic plasticity-related gene downregulation [71]. The study also found a significant decrease in the nuclear histone deacetylase 4 (HDAC4) expression, which might relate to MiR-29a enriched in MSC-Exos with a specific action on HDAC4 [71]. Despite MSC-Exos could enter the brain, there were limited amounts of MSC-Exos accumulating in the brain compared to the spleen and liver [67,69]. To enhance brain targeting ability, various methods have been adopted, such as peptide modification and gene transfection [72,73,74]. The modification of a CNS-specific rabies viral glycoprotein (RVG) peptide on MSC-Exos improved the brain-targeting efficiency via interaction between RVG and acetylcholine receptors on neuronal cells [72]. The MSC-Exos conjugated with a cyclo (Arg-Gly-Asp-D-Tyr-Lys) peptide [c(RGDyK)] with a high targeting ability to integrin α_v_β_3_ and the cRGD-MSC-Exos showed a higher accumulation in the brain than the MSC-Exos without modification [73]. Moreover, the MSC-Exos isolated from MSCs transfected with CXCR4 and tumor necrosis factor-related apoptosis-inducing ligand (TRAIL) enhanced the efficacy of MSC-Exos in combination therapy with carboplatin [74]. In the study, CXCR4 played a key role in adhesion molecule expression, and homing ability, and thus enhanced the delivery of MSC-Exos to the brain lesions [74].

MSC-EVs can carry nanoparticles for combination therapy for brain diseases. The angiopep-2 (TFFYGGSRGKRNNFKTEEY, ANG) peptide-decorated MSC-Exos (ANG-EXO) could specifically bind with low-density lipoprotein receptor protein 1 (LRP1) expressing on the brain capillary endothelial and glioma cells, which mediated drug delivery across the BBB (Figure 5B) [75]. siRNA of GPX4 (glutathione peroxidase 4, a key regulator of ferroptosis) was loaded into the ANG-EXO by electroporation, and the thus-formed ANG-EXO-siGPX4 conjugated with the brequinar-loaded magnetic nanoparticles (MNPs@BQR) (Figure 5A) [75]. The MNPs@BQR@ANG-EXO-siGPX4 could penetrate through the BBB together and induce ferroptosis by the combined action of siGPX4 and MNPs@BQR (Figure 5C,D) [75]. Brequinar is an inhibitor of dihydroorotate dehydrogenase that is a therapeutic target for ferroptosis.

#### 3.1.2. Neural Stem Cells (NSCs) and Neural Progenitor Cells (NPCs)

NSCs, derived from the neuroepithelial cells of the neural tube, maintain the capacity for self-renewal and generation of neurons, astrocytes, and oligodendrocytes throughout their whole lives [76]. The NSC-EVs with inherent property to traverse the BBB can regulate the pathological microenvironment and signaling pathways, offering the potential of treating brain diseases [77]. For example, NSC-EVs have been reported for thromboembolic stroke treatment, which can significantly reduce nerve damage in the mouse model of thromboembolic stroke, thus reducing behavioral and motor function deficits [78]. The engineered NSC-EVs showed higher BBB penetration efficiency compared with naive NSC-EVs [78]. In multiple sclerosis lesions, the oligodendrocyte progenitor cells with high expression of platelet-derived growth factor receptor α accumulated in the demyelinated areas [78]. The NSC-EVs modified with platelet-derived growth factor-A enhanced the efficiency of targeted delivery to the CNS and improved the therapeutic effect of Bryostatin-1 encapsulated in the EVs [78].

NPCs, differentiated from NSCs, can give rise to the glial and neuronal cells that populate the CNS [79] and are an ideal source for the production of EVs as therapeutic platforms for brain delivery. EVs isolated from ReN cells, a kind of neural progenitor cell line, were conjugated with c(RGDyK) peptide to overcome the BBTB and enhance glioblastoma-targeted delivery [80]. It was found that short-burst of radiation increased the accumulation of the RGD-EVs in the glioblastoma [80]. Moreover, the EVs with small interfering RNA (siRNA) against programmed cell death ligand-1 (PD-L1) (siPDL1) attachment on its membrane, termed RGD-EV:siPDL1, could decrease the radiation-induced PD-L1 expression on tumor-associated myeloid cells and tumor cells, revealing the potential of combination therapy of NPC-EVs and short-burst radiation [80].

#### 3.1.3. Other Stem Cells

Endometrial stem cells (EnSCs) in the basal and functional layers of the human endometrium involve in the remodeling of the tissue [81]. The EVs derived from EnSCs with anti-apoptotic, pro-angiogenic, and immunomodulatory functions can be used as delivery platforms [81]. In a previous study, the exosomes harvested from human EnSCs, defined as hEnSCs-EXOs, were applied to incorporate curcumin to alleviate the symptoms of Parkinson’s disease and reduced neuroinflammation [82].

Embryonic stem cells (ESCs) are pluripotent stem cells derived from the inner cell mass of blastocysts and characterized by unlimited proliferation, self-renewal, and multidirectional differentiated ability [83]. ESCs can be induced to differentiate into almost all cell types in the body and produce EVs on a large scale, which is an advantage for clinical application [83]. The tumor-targeting ESC-exos were prepared by conjugating ESC-exos with c(RGDyk) peptide for brain delivery, showing an efficient accumulation in the glioma site [84].

### 3.2. Immune Cells

Immune cell-deriving EVs play a key role in a lot of physiological and pathological processes and have promising applications for drug delivery [85]. Immature dendritic cells can produce large amounts of EVs lacking T cell activators (e.g., MHC-II and costimulatory molecules), and they can serve as a useful cell source of EVs for treating neurodegenerative diseases [86]. The immature dendric cell-derived exosomes were modified with RVG (defined REXO) and thus used as a shell to wrap the amphiphilic polymer nanoparticles containing curcumin and siRNA targeting *SNCA* (siSNCA) for Parkinson’s disease treatment [87]. This delivery system co-delivered curcumin and siSNCA to the site of brain lesions and exerted therapeutic effects [87]. In another study, the RVG peptide-modified exosomes were used to deliver anti-α-synuclein shRNA minicircles, with extended circulation time in vivo and specific delivery to the brain [88].

The EVs derived from macrophages could traverse the BBB by interacting with adhesion molecules and certain cell surface ligands like intercellular adhesion molecule 1 (ICAM-1), integrin white corpuscle function-associated matter 1, and carbohydrate-binding C-type glycoprotein receptors [89]. For instance, the AS1411 aptamer-modified macrophage exosomes (Ex-A) were explored for efficient BBB penetration and glioblastoma sonodynamic therapy [90]. The Ex-A loaded with catalase and sonosensitizer ICG had a high tumor-targeted ability and efficient BBB penetration, as well as a high accumulation in the tumor site, yielding enhanced sonodynamic therapy of glioblastoma [90]. Based on the fact that macrophage-derived EVs preferred to accumulate in the inflamed brain sites, the macrophage exosomes were used to carry brain-derived neurotrophic factors to the intracranial inflammatory lesion [91]. Besides, the biomimetic macrophage-derived EVs were developed to deliver nanomedicine into the brain; the cRGD-modified exosome membranes were incorporated with the micelles that loaded with panobinostat and p53-induced protein phosphatase 1 (PPM1D) siRNA, and the composite system showed significant accumulation in the brain and targeting ability to tumor cells (Figure 6) [92]. Considering the limited drug-loading capacity of natural EVs, the combination of the EV membrane with other nanocarriers is a potential drug delivery strategy.

As described in the section above, NEs have been extensively researched as brain disease therapeutic platforms due to their inflammation-homing and BBB penetration ability [17]. The NE-derived EVs have been also used for brain delivery; for instance, the NE EVs loaded with doxorubicin could efficiently inhibit tumor growth and prolong the survival time in a glioblastoma mouse model [93].

### 3.3. Tumor Cells

On account of the organ tropism, the EVs derived from tumor cells have the tumor-homing ability [94]. Therefore, tumor cell-derived EVs are the potential tools for intracranial tumor and brain inflammation therapy.

Apoptotic bodies (ABs) are a class of EVs secreted by apoptotic cells. Compared with other EVs, ABs show high cargo-loading capacity, because the apoptotic cells actively package biomolecules into vesicles of ABs [91]. The highly metastatic tumor cells internalizing anti-TNF-α antisense oligonucleotide complexed with a cationic konjac glucomannan (cKGM) were used to generate small ABs (sABs); the complexed cKGM could mediate the internalization of the complex by the tumor cells via mannose receptors on the cell membranes (Figure 7A) [95]. The sABs efficiently delivered anti-TNF-α antisense oligonucleotide (ASO) into the brain in a Parkinson’s disease animal model; the brain delivery process involved the transcytosis through the BBB mediated by CD44v6 expressed on the membrane of ABs (Figure 7B) [95]. The delivery system successfully reduced brain inflammation and ameliorated Parkinson’s disease symptoms (Figure 7C) [95].

However, accumulating evidence has demonstrated that tumor cell-derived EVs are involved in tumor progression and remodeling of the immune microenvironment [96,97]. To avoid the potential pro-tumor effect, several approaches were applied to remove the natural contents of tumor-cell-derived EVs. For example, ultrasonic fragmentation was used to remove the inherited DNA and proteins [98]. The resulting reassembly exosomes (R-EXO) retained the natural homologous targeting ability to penetrate the BBB and carried temozolomide and dihydrotanshinone to the tumor site and had potent therapeutic effects on glioma [98].

### 3.4. Plants

Plant-derived EVs have the advantages of low cost, high productivity, and high stability [99]. Plant-derived EVs contain the innate active components secreted by the source plants, which have biological functions such as anti-inflammatory, antioxidant, and antitumor [99]. Therefore, plant-derived EVs can serve as potential drug carriers and therapeutic agents.

A previous study showed that EVs derived from *Momordica. charantia* could penetrate the BBB and accumulate in the brain parenchyma [100]. The grapefruit-derived EVs with the pH-sensitive doxorubicin-loaded heparin-based nanoparticles patching on their membranes were designed for glioma therapy [101]. In addition, the doxorubicin-loaded nanoparticles were modified by RGD peptide to further enhance tumor delivery capabilities [101]. This complex platform achieved high accumulation at the glioma sites due to the natural BBB-penetrating ability of grapefruit-derived EVs and α_v_β_3_ receptor-mediated transcytosis and membrane fusion [101].

### 3.5. Other Sources

The safety of EVs derived from blood has been demonstrated by blood transfusions [102]. A previous experiment demonstrated that blood EVs possessed natural tropism to penetrate the BBB, based on the transferrin receptor (TfR) expressed on their surfaces [103]. It is great potential for blood EVs for drug delivery for brain disease treatment. Loaded with metformin and cytoplasmic phospholipase A2 siRNA (sicPLA2), the blood EVs could efficiently deliver these therapeutic agents to glioblastoma cells and show high efficacy in a patient-derived glioblastoma orthotopic xenograft model [104].

The BBB-penetrating abilities of exosomes isolated from 4 types of cells, including brain glioblastoma U-87 MG cells, and brain endothelial bEND.3 cells, neuroectodermal tumor PFSK-1 cells, and glioblastoma A-172 cells have been evaluated [105]. The bEND.3 exosomes showed a higher delivery efficiency of doxorubicin and paclitaxel than other exosomes, due to the brain-homing activity mediated by the high expression of CD63 on their membranes, which is a tetraspanin protein mediating cell-cell communication between astrocytes and cortical neurons [105]. Another study also demonstrated that the rat cerebral endothelial cell-derived EVs could penetrate the BBB and be efficiently internalized by neurons and other parenchymal cells [106]. Therefore, brain endothelial cell-derived EVs have the potential as brain drug delivery carriers.

Milk is an abundant source of EVs, and the milk EVs can remain stable in the gastrointestinal tract, thus protecting the cargo from degradation by low pH and enzymes [107]. After oral administration, the milk-derived exosomes loaded with curcumin showed long circulation time and high accumulation of curcumin in the brain, which indicated the potential capacity of milk-derived EVs to transport bioactive components in the brain [108]. It provides a very attractive method for brain delivery via a non-invasive administration route.

**Table 2 pharmaceutics-15-01257-t002:** EV-mediated BBB-crossing delivery.

EV Source	Surface Modification	Disease Model	Cargo	Administration Method	In Vitro/In Vivo Improvement	Refs.
Mesenchymal stromal cells	cRGD	Cerebral ischemia	Curcumin	IV	Showed better anti-inflammatory effects than free curcumin	[73]
Mesenchymal stem cells	RVG	Alzheimer’s disease	NA	IV	Reduced plaque deposition and Aβ accumulation and improved learning and memory capabilities	[71]
Adipose-derived stem cells	NA	Ischemic stroke	miRNA-126	IV	Enhanced neurogenesis, inhibited neuroinflammation, and improved functional recovery	[109]
Mesenchymal stem cells	CXCR4	Brain metastasis of breast cancer	TRAIL	IV	Improved EVs delivery across the BBB and exerted cooperative therapeutic effects with carboplatin	[74]
Human mesenchymal stem cells	ANG	Glioblastoma	Magnetic nanoparticles loaded with brequinar	IV	Enabled the cargo transport through the BBB and enhanced ferroptosis induced by magnetic nanoparticles	[75]
Human umbilical cord mesenchymal stem cells	NA	Parkinson’s Disease	NA	IV	Reached the substantia nigra and decreased dopaminergic neuron loss and therefore upregulating dopamine levels in the striatum	[110]
Neural stem cells	PDGF-A	Experimental autoimmune encephalomyelitis	Bryostatin-1	IV	Enhanced the targeting abilities toward brain lesions and showed excellent therapeutic capabilities of reducing neuroinflammation with low dosages of Bryostatin-1	[78]
Neural progenitor cells	RGD-4C	Ischemic stroke	NA	IV	A set of seven miRNAs incorporated in EV_ReN_ inhibited the MAPK signaling pathway; significantly improved the targeting and therapeutic efficacy	[111]
Neural progenitor cells	cRGD	Glioblastoma	siRNA against PD-L1	IV	Decreased radiation-induced PD-L1 expression on tumor-associated myeloid cells and tumor cells as well as the upregulation of CD8^+^ cytotoxic T cells	[80]
Human endometrial stem cells	NA	Parkinson’s Disease	Curcumin	IV	Suppressed α-synuclein aggregation and neural cell apoptosis	[82]
Embryonic stem cells	cRGD	Glioblastoma	Paclitaxel	IV	Improved the therapeutic efficacy of paclitaxel via superior glioblastoma-targeting capacity	[84]
Dendric cells	RVG	Parkinson’s Disease	Anti-α-synuclein shRNA	IV	Achieved long-term downregulation of target genes, inhibition of dopaminergic cell death and movement abnormalities	[88]
Immature dendric cells	RVG	Parkinson’s Disease	Curcumin; siRNA targeting *SNCA*	IV	Enhanced drug delivery via modification, improved α-synuclein clearance and neuronal recovery and cleared aberrant immune activation	[87]
Naive macrophages	NA	Brain inflammation	Brain-derived neurotrophic factor	IV	Interacted with endothelial ICAM-1 thus mediating the migration across the BBB, improved the cargo accumulation in the brain	[91]
Macrophages	RGE	Glioma	Superparamagnetic iron oxide nanoparticles and curcumin	IV	Showed significant antitumor effects synchronized with the MRI ability in vivo	[112]
Macrophages	NA	Alzheimer’s disease	Curcumin	IV	Activated AKT and inhibited GSK-3β, resulting in the reduction of Tau protein phosphorylation; improved cognitive function in mice model	[113]
Macrophages	NA	Glioma	Boron-containing carbon dots	IV	Improved the therapeutic effect of boron neutron capture therapy due to the high accumulation of boron-containing carbon dots via exosome delivery in mice brain	[114]
Macrophages	cRGD	Diffuse intrinsic pontine glioma	Panobinostat; anti PPM1D siRNA	IV	Achieved excellent tumor growth inhibition ascribed to the high BBB penetration efficiency	[92]
M1-like macrophages	CPPO; Ce6	Glioblastoma multiforme	Banoxantrone	IV	Induced M2 polarization to M1 for the immunomodulation of TME; combined with chemiexcited photodynamic therapy after accumulating in brain tumors	[115]
Macrophages	AS1411	Glioblastoma	ICG; catalase	IV	Realized the high accumulation in the tumor site and enhanced the sonodynamic therapy of glioblastoma	[90]
Neutrophils	NA	Glioblastoma	Doxorubicin	IV	Penetrated the BBB and targeted tumor in response to inflammation; inhibited tumor growth and extended survival time in a glioblastoma mouse model	[93]
Tumor cells	NA	Parkinson’s Disease	ASO; cKGM	IV	Efficiently delivered cargos into the brain, enhanced by the transcytosis through BMECs, which is mediated by CD44v6 expressed on the membrane; reduced brain inflammation and ameliorated Parkinson’s disease symptoms	[95]
Tumor cells	NA	Glioblastoma multiforme	Dihydrotanshinone; temozolomide	IV	Overcome the drug resistance of temozolomide and triggered the immune response	[98]
Momordica charantia	NA	Glioma	NA	IV	Inhibited glioma progression via the regulation of PI3K/AKT pathway	[100]
Grapefruit	NA	Glioma	Doxorubicin-loaded heparin-based nanoparticles	IV	Achieved high-abundance accumulation and brain tumor uptake at glioma sites by α_v_β_3_ receptor-mediated transcytosis and membrane fusion	[101]
Cerebral endothelial cells	NA	Ischemic stroke	NA	IV	Penetrated the BBB, efficiently internalized by neurons and other parenchymal cells, and improved fibrinolysis and thrombolysis combined with tPA	[106]
Blood	NA	Parkinson’s Disease	Dopamine	IV	Penetrated the BBB based on the transferrin receptor expressed on the surfaces; reduced systemic toxicity of dopamine	[103]
Blood	NA	Glioblastoma	Metformin; cPLA2 siRNA	IV	Showed great efficacy in the patient-derived glioblastoma orthotopic xenograft model by improving glioblastoma cell uptake via high expression of PTRF	[104]
HEK 293T	T7 peptide	Glioblastoma	Antisense miRNA oligonucleotides against miR-21	IV	Showed more efficient delivery of cargoes than RVG-decorated exosomes and inhibited tumor growth by increasing the expression of PDCD4 and PTEN	[63]
HEK 293T	RVG	The chronic unpredictable stress model	CircDYM	IV	Suppressed microglia activation by bounding to TAF1; reduced the infiltration of peripheral immune cells, thereby attenuating depressive-like behaviors	[60]
HEK 293T	ANG and TAT peptide	Glioma	Doxorubicin	IV	Improved the efficiency to penetrate the BBB and penetrate the tumor, resulting in the high therapeutic efficacy of the chemotherapeutic drug	[62]

cRGD, cyclo (Arg-Gly-Asp-D-Tyr-Lys); IV, intravenous injection; RVG, rabies virus glycoprotein; CXCR4, C-X-C chemokine receptor type 4; TRAIL, tumor necrosis factor-related apoptosis-inducing ligand; ANG, angiopep-2; PDGF-A, platelet derived growth factor subunit A; RGD-4C, arginine-glycine-aspartic acid (RGD)-4C peptide (ACDCRGDCFC); EV_ReN_, ReN cell-derived EVs; MAPK, mitogen-activated protein kinase; PD-L1, programmed death ligand-1; *SNCA*, the alpha-synuclein gene; ICAM-1, intercellular adhesion molecule 1; RGE, neuropilin-1-targeted peptide; MRI, magnetic resonance imaging; PPM1D, p53-induced protein phosphatase 1; CPPO, hydrophobic bis(2,4,5-trichloro-6-carbopentoxyphenyl) oxalate; Ce6, chlorin e6; TME, tumor microenvironment; ICG, indocyanine green; ASO, anti-TNF-α antisense oligonucleotide; cKGM, cationic konjac glucomannan; BMECs, brain microvascular endothelial cells; tPA, tissue plasminogen activator; cPLA2, cytoplasmic phospholipase A2; PTRF, polymerase 1 and transcript release factor; T7 peptide, transferrin receptor-binding peptide; PDCD4, programmed cell death protein 4; PTEN, phosphatase and tensin homologue; CircDYM, a homologous human and mouse circRNA derived from exons 4, 5 and 6 of the DYM gene; TAF1, TATA-box binding protein associated factor 1; TAT, trans-activator of transcription.

## 4. Cell Membrane-Cloaked Nanoparticles for BBB-Crossing Delivery

Cell membrane-cloaked nanoparticles consist of a core and an outer layer membrane [116]. The core can carry therapeutic agents with different biological and therapeutic functionalities [116], while a cell membrane layer contains the inherited cell surface proteins that can prolong blood circulation time and enhance tumor accumulation [116,117]. Cell membranes carry the lipids, protein, and carbohydrates originating from the source cells [118], and can be coated onto the nanoparticle surface via three main approaches such as sonication, extrusion, or microfluidic mixing [117]. The use of cell membranes from various sources for brain delivery is summarized in Table 3.

### 4.1. Red Blood Cells (RBCs)

RBCs are responsible for oxygen transport throughout the body and have been widely studied as carriers for various bioactive compounds [119]. The membrane proteins (e.g., CD47) on the RBC are typically preserved, which is important for long-term systemic circulation [119,120]. RBC membrane-camouflaged nanoparticles are characterized by long-circulating time, reduced reticuloendothelial system uptake, and non-immunogenicity, and are emerging as promising nanocarriers [117,119,121].

RBC membranes were camouflaged on the protein superstructures to facilitate the delivery across the BBB [122]. The RBC membrane-coated nanoparticles (RBC@Hb@GOx NPs) loaded with hemoglobin (Hb) and glucose oxidase (GOx) were developed for treating glioblastoma (Figure 8A,B) [122]. They could penetrate the BBB and BBTB and passively accumulate in the brain tumors taking advantage of low immunogenicity and long circulation of the RBC membrane (Figure 8C), thus producing reactive oxygen species (ROS) and efficiently inhibiting glioblastoma growth (Figure 8D) [122].

To further achieve BBB penetration, surface modifications can be applied to the erythrocyte membrane. For example, the RBC membrane-camouflaged PLGA nanoparticles were functionalized with T807 molecules, which could specifically bind to phosphorylation tau mainly located in the neurons, thus further enhancing curcumin accumulation in the brain [123]. After systemic administration, T807@RBC@Cur NPs loaded with curcumin could effectively alleviate Alzheimer’s disease progression [123]. Another example is that the dual-targeted nanoerythrocytes engineered with RVG29 peptide (with a long arm) and MG1 peptide (with a short arm) were developed (termed NEMR) to enhance brain delivery and alleviate the MCAO and experimental autoimmune encephalomyelitis (EAE) [124]. The engineered NEMR could penetrate the BBB with the aid of RVG29, and then RVG29 was shed (termed NEM) in response to ROS to promote MG1 exposure and the subsequent targeting M1 microglia, thereby reprogramming M1 microglia (proinflammatory) to M2 phenotype (anti-inflammatory) via heme-induced up-regulation of heme oxygenase-1 (HO-1) [124].

### 4.2. Immune Cells

#### 4.2.1. Natural Killer (NK) Cells

NK cells are a crucial group of lymphocytes in the innate immune system [125]. NK cell membrane proteins (e.g., lymphocyte function-associated antigen 1 (LFA-1) and VLA-4) can interact with endothelial cell adhesion molecules (e.g., ICAM-1 and VCAM-1), and then trigger an intracellular signaling cascade associated with tight junction (TJ) disruption and actin cytoskeleton reorganization (Figure 9E) [126]. Therefore, NK cell membrane proteins could serve as TJ modulators for enhancing BBB permeability (Figure 9B,F,G) [126]. Meanwhile, NK cell membrane coating enabled the nanoparticles to target glioma cells via the membrane proteins (e.g., NKG2D and DNAM-1) (Figure 9C,D) [126]. Via this approach, NK cell membrane-coated aggregation-induced emission (AIE)-active polymeric nanoendoskeleton (NK@AIEdots) could deliver the cargo through the BBB and highly accumulate in the brain tumor (Figure 9A) [126]. By intravascular injection in an orthotopic U87 glioma model, NK@AIEdots effectively inhibited the glioma growth via NIR-II-guided skull imaging and photothermal therapy [126].

#### 4.2.2. Macrophages

Macrophage membranes have also been utilized for coating on the surface of nanocarriers, thus endowing them with low immunogenicity, prolonged half-life, and precise brain targeting capability [127]. Specific integrins (e.g., α4 and β1 integrins) on macrophage membranes can bind the VCAM-1 overexpressed on the cancer cell [128,129], thus enabling macrophage membrane-coated nanoparticles for effective glioma targeting delivery [130]. Macrophage membranes were used to coat poly(N-vinyl caprolactam) nanogels embedded with manganese dioxide (MnO_2_) and cisplatin (M@Pt/MnO_2_@PVCL NGs), enabling them to penetrate the BBB for chemotherapy/chemodynamic therapy in a mouse model of orthotopic glioma [131]. In another example, the MnO_2_ nanospheres incorporating fingolimod were disguised with macrophage membranes (Ma@MnO_2_@FTY NPs) to salvage the ischemic penumbra [127]. The macrophage membrane proteins (CD44 and CD11b) endowed the nanocarriers with inflammation-oriented chemotactic ability via ICAM-1 and P-selectin overexpressed on the stressed vascular endothelial cells [127]. The Ma@MnO_2_@FTY NPs accumulated in the ischemic region, and then reversed the proinflammatory microenvironment and reinforced neuroprotective effects in a rat transient middle cerebral artery occlusion/reperfusion (tMCAO/R) model [127].

Genetic engineering is a promising approach for cell membrane modification. For example, the PLGA nanoparticles coated with programmed cell death-1 (PD-1)-engineered macrophage membrane were constructed (PD-1-MM@PLGA/RAPA NPs) to facilitate immune checkpoint blockade combined with chemotherapy (e.g., Rapamycin) for glioblastoma treatment (Figure 10A) [132]. PD-1-MM@PLGA/RAPA NPs could permeate the BBB and subsequently accumulate in the glioblastoma microenvironment (Figure 10B,C) via macrophage chemoattractants secreted from glioblastoma cells, and further block the PD-1/PD-L1 axis (Figure 10D,E) [132].

#### 4.2.3. Neutrophils

There is no simple protocol to generate large numbers of neutrophils for therapeutic purposes because neutrophils have a very short circulatory half-life and are difficult to cultivate and retain their inherited functions. Alternatively, employing the neutrophil membrane for coating nanoparticles can be an option [8]. In a study, the neutrophil membrane-coated mesoporous Prussian blue nanozymes (MPBzyme@NCMs) were designed for targeting the damaged brain via the interaction between ICAM-1 on inflamed brain microvascular endothelial cells and LFA-1 and macrophage-1 antigen (Mac-1) overexpressed on neutrophils [133]. MPBzyme@NCM treatment on ischemic stroke elicited M2 microglia polarization, reduced neutrophil recruitment, and protected against neuronal damage in a mouse model of transient middle cerebral artery occlusion (tMCAO) [133]. Similarly, the neutrophil membrane-derived nanovesicles loading Resolvin D2 (RvD2) were developed to protect against brain damage and inhibit neuroinflammation during ischemic stroke [134].

### 4.3. Tumor Cells

Cancer cell membrane coating has been actively explored for cancer drug delivery by leveraging the homotypic recognition characteristics of cancer cells [117]. Inspired by the brain metastasis of breast cancer, lung cancer, and melanoma, these tumor cell membranes were used to coat the nanoparticle to mediate the BBB-penetrating delivery [135,136]. As an example, the glioma cell membrane-coated nanoparticle (M@HLPC) was prepared via the self-assembly of hemoglobin, lactate oxidase, bis[2,4,5-trichloro-6-(pentyloxycarbonyl)phenyl] oxalate (CPPO), and chlorin e6 (Ce6) [135]. M@HLPC exhibited high glioma accumulation due to homologous targeting, in which lactate oxidase converted lactate into pyruvic acid and H_2_O_2_, which subsequently reacted with CPPO to generate chemical energy for activating the photosensitizer Ce6, thus performing tumor-killing functions [135]. In another example, indocyanine green was loaded into the polymeric nanoparticles that were then coated with various brain metastatic tumor cell membranes (e.g., B16F10 cell or 4T1 cell) to fabricate the biomimetic nanoparticles (B16-PCL-ICG NPs or 4T1-PCL-ICG NPs) [136]. Intravenous injection of these nanoparticles combined with photothermal therapy substantially inhibited glioma growth [136]. Moreover, the PC@siRNA nanocomplexes were prepared by electrostatic interaction between polyethyleneimine xanthate (copper chelating agent) and siRNA Bcl-2 (siBcl-2); the nanocomplexes were coated with citraconic anhydride grafted poly-lysine (PLL-CA), and further merged with B16F10 melanoma cell membrane (MPC@siRNA) (Figure 11A) [137]. MPC@siRNA exhibited prolonged circulation, increased BBB permeability, and enhanced glioblastoma accumulation due to reversible BBB opening mediated by MPC@siRNA (Figure 11B) and homologous targeting (Figure 11C,D) [137].

Cancer cell membrane coating has also been used for ischemic stroke treatment. The overexpressed CD138 and VCAM-1 on the 4T1 cell membrane can interact with CD31 and VLA-4 on various cerebral cells (e.g., platelets, vascular endothelial cells, and leukocytes), thus facilitating the cargos to penetrate the BBB and target the cerebral ischemic lesions [138]. 4T1 cell membranes have been used to coat a pH-sensitive polymer of methoxy poly(ethylene glycol)-block-poly(2-diisopropyl methacrylate) (PEG−PDPA) loaded with succinobucol, termed MPP/SCB [138]. Succinobucol was released from the MPP/SCB in response to the intracellular acidic environment and then exhibited appreciable neuroprotective effects on tMCAO rats [138].

### 4.4. Stem Cells

The NSC membrane was engineered to overexpress CXCR4, which substantially interacted with SDF-1 enriched in the ischemic region, and thus enhanced the delivery of the payload to the ischemic area [139]. Glyburide, an antistroke agent, was incorporated into the PLGA nanoparticle cores, followed by coating with the NSC membrane engineered with CXCR4; this strategy was used for the stroke-targeting delivery and stroke treatment due to the interaction of SDF-1-CXCR4 axis [139].

### 4.5. Bacteria

The Gram-negative bacterial membrane can interact with gp96 on BBB endothelial cells via its outer membrane protein A (OmpA) component and has a potential application for brain-targeted drug delivery [140]. The LPS-free outer membrane vesicles (OMV) from *Escherichia coli* K1, were used to encapsulate the doxorubicin-loaded PLGA nanoparticles to treat breast cancer brain metastases; the brain-targeted delivery was achieved via the OmpA-gp96 interaction-mediated transcytosis [140].

### 4.6. Hybrid Membranes

Hybrid membrane frameworks were prepared using various types of cell membranes, including tumor cell-DC [141], tumor cell-RBC [142], and macrophage-neutrophil [143]. The hybrid membranes from C6 cells and DCs were constructed to disguise the docetaxel-loaded nanosuspensions (DNS-[C6&DC]m), and the thus-formed system had the tumor cell membrane-mediating homologous targeting characteristics and the antigen-presenting properties associated with DCs [141]. The DNS-[C6&DC]m had a prolonged circulation time and could penetrate the BBB and BBTB, deliver docetaxel to the brain tumor and induce anti-tumor immune responses [141]. Similarly, the hybrid membranes derived from human glioma cells (U251) and RBCs were applied to deliver isoliquiritigenin for prolonged blood circulation and homotypic targeting [142]. In another study, the rapamycin-loaded PLGA nanoparticles were camouflaged with the hybrid cell membrane from neutrophil and macrophage (NMm-PLGA/RAPA NPs) and exhibited BBB permeability and inflammatory chemotactic activities [143].

**Table 3 pharmaceutics-15-01257-t003:** Cell membrane-cloaked nanoparticles for BBB-crossing delivery.

Cell Types	Surface Functionalization	Targets	Disease Model	Cargo	Loading Mechanism	Administration Method	Release Mechanism	In Vitro/In Vivo Improvement	Ref.
RBC membrane	NA	NA	Orthotopic gliomas (U87 cells)	Hb@GOx NPs	Sonication	IV	In situ H_2_O_2_ causes cell membrane destruction	Exhibited notable tumor accumulation and inhibited the growth of GBM	[122]
RBC membrane	T807	Phosphorylation tau	AD	Curcumin	Sonication and extrusion	IV	NA	Increased curcumin accumulation in the brain and alleviated AD progression	[123]
Nanoerythrocyte membrane	MG1 peptide/RVG29 peptide	M1 microglia	MCAO and EAE	NEMR	Extrusion	IV	ROS	Penetrated the BBB, targeted M1 microglia, re-educated M1 to M2 microglia, suppressed inflammation, and enhanced neuroprotection	[124]
NK cell membrane	NA	NA	Orthotopic gliomas (U87 cells)	AIEdots	Extrusion	IV	NA	Exhibited higher tumor accumulation, and inhibited the growth of glioma	[126]
Macrophage membrane (RAW264.7 cells)	NA	NA	Orthotopic gliomas (C6 cells)	Pt/MnO_2_@PVCL NGs	Extrusion	IV	ROS/pH	Exhibited BBB penetrating and glioma targeting ability for MR imaging-guided chemotherapy/CDT	[131]
Macrophage membrane (rat peritoneal macrophages)	NA	NA	tMCAO/R model	MnO_2_@FTY NPs	Extrusion	IV	ROS/pH	Accumulated in the ischemic region, repolarized M1 microglia to M2 microglia, reduced oxidative stress, reversed the proinflammatory microenvironment, and reinforced neuroprotective effects	[127]
Macrophage membrane (RAW264.7 cells)	PD-1	PD-L1	Orthotopic gliomas (C6 cells)	PLGA/RAPA NPs	Extrusion	IV	NA	Enhanced the BBB penetration, blocked the PD-1/PD-L1 axis, and exhibited anti-GBM efficacy	[132]
Neutrophil membrane (HL60 cells)	NA	NA	tMCAOmodel	MPBzyme	Extrusion	IV	NA	Elicited M2 microglia polarization, reduced neutrophil recruitment, and protected against neuronal damage	[133]
Neutrophil membrane (HL60 cells)	NA	NA	MCAO model	Resolvin D2	Sonication	IV	NA	Targeted inflamed brain endothelium, and mitigated neuroinflammation	[134]
Tumor cell membrane (U251 glioma cells or human tumor tissue)	NA	NA	Orthotopic glioma model (U251 cells)/patient-derived xenograft tumor models (human tumor tissue)	HLPC	Extrusion	IV	NA	Enhanced glioma-targeting efficacy, induced cell apoptosis, and exhibited the antitumor effect	[135]
Tumor cell membrane (B16F10 cells or 4T1 cells)	NA	NA	Orthotopic glioma model (U87 cells)	PCL-ICG NPs	Sonication	IV	NA	Improved the long-circulating capacity, traversed the BBB, accumulated in glioma cells, and inhibited glioma growth.	[136]
Tumor cell membrane (B16F10 cells)	NA	NA	Local subcutaneous melanoma model (B16F10 cells)/orthotopic GBM model (U87MG, GL261 cells)	PC@siRNA	Sonication	IV	pH	Promoted BBB penetration and GBM accumulation; inhibited the growth of orthotopic GBM tumors and subcutaneous melanoma tumor	[137]
Tumor cell membrane (4T1 cells)	NA	NA	tMCAO model	PP/SCB	Extrusion	IV	pH	Penetrated the BBB, targeted the ischemic inflammation lesions, reduced the infarct volume, enhanced microvascular reperfusion, and promoted neuroprotective effects	[138]
NSC membrane	CXCR4	SDF-1	MCAO	Gly@PLGA NPs	Extrusion	IV	NA	Augmented the efficacy of glyburide, promoted the accumulation of nanoparticles in the ischemic region	[139]
LPS-free OMV from EC-K1	NA	NA	Brain metastasis model (231Br cells)	DOX@PLGA NPs	Extrusion	IV	NA	Prolonged circulation, improved BBB penetration, enhanced brain-targeted ability, and lengthened the survival of breast cancer brain metastases	[140]
Hybrid membrane (C6 cells and DCs)	NA	NA	Intracranial glioma model (C6 cells)	DNS	Sonication	IV	NA	Prolonged the blood circulation time, penetrated the BBB and BBTB, delivered docetaxel to the tumor site, and enhanced the anti-tumor immune response	[141]
Hybrid membrane (RBC and U251 cells)	NA	NA	NA	Isoliquiritigenin	Sonication and extrusion	NA	NA	Inhibited U251 cell migration and promoted U251 cell apoptosis	[142]
Hybrid membrane (RAW 264.7 cells and neutrophils)	NA	NA	Mouse brain inflammatory model/glioma (C6 cells)	PLGA/RAPA NPs	Sonication and extrusion	IV	NA	Recognized the chemotactic stimuli, transported across BBB, and showed the antitumor effect	[143]

Hb, hemoglobin; Gox, glucose oxidase; IV, intravenous injection; GBM, glioblastoma; T807, AV-1451; AD, Alzheimer’s disease; MCAO, middle cerebral artery occlusion; EAE, experimental autoimmune encephalomyelitis; NEMR, an MG1 peptide and RVG29 peptide engineered nanoerythrocyte; ROS, reactive oxygen species; AIEdots, nanorobots with aggregation-induced emission characteristics; PVCL, poly(N-vinylcaprolactam); NGs, nanogels; MR imaging, magnetic resonance imaging; CDT, chemodynamic therapy; tMCAO/R, transient middle cerebral artery occlusion/reperfusion; FTY, fingolimod; PD-1, programmed cell death-1; PD-L1, programmed cell death-ligand 1; PLGA, poly(lactic-co-glycolic acid); RAPA, rapamycin; tMCAO, transient middle cerebral artery occlusion; MPBzyme, mesoporous Prussian blue nanozyme; MCAO, the middle cerebral artery occlusion; HLPC, Hb, lactate oxidase, CPPO, and Ce6; PCL, poly(caprolactone); ICG, indocyanine green; PC, citraconic anhydride grafted poly-lysine; PP/SCB, pH-sensitive polymeric nanoparticles of succinobucol; CXCR4, C-X-C chemokine receptor type 4; SDF-1, stromal cell-derived factor 1; OMV, outer membrane vesicles; EC-K1, Gram-negative Escherichia coli K1; DOX, doxorubicin; DNS, docetaxel nanosuspensions; DCs, dendritic cells.

## 5. Conclusions and Future Perspectives

Cell- and cell-derivative-based delivery systems have been demonstrated with promising preclinical results in overcoming the formidable BBB challenge. Yet, continual efforts must be made to promote their translation from bench to bedside.

First of all, there is still a bottleneck issue to achieving large-scale production of carrier cells or exosomes. For example, it is difficult to produce large numbers of neutrophils, since these short-lived cells are difficult to cultivate and can only be isolated from peripheral blood [144,145]. A possible solution is to inject a tailored nano-drug into the blood, where they can get internalized by circulating neutrophils [20,21]. However, the injected nano-drug may also be rapidly recognized and cleared by the mononuclear phagocyte system [146]. Moreover, there are other options, for example, choosing appropriate sources, developing new enrichment methods, and searching for mimetic substitutes. The plant-derived EVs are an economical alternative with widespread sources and high cost-efficiency (about 300-fold lower cost/yield ratio to mammalian-derived EVs) [11,147]. Various new enrichment methods including microfluidic filtering, contact-free sorting, and immunoaffinity enrichment have been developed to improve the yield and purity of EVs from complex biological fluids [148]. In addition, EV-mimetic nanoparticles can be developed as alternatives, which include nanovesicles generated by subjecting cells to serial extrusion through filters, cell membrane nanovesicles generated by sonication-assisted reassembly of repeatedly freeze-thawed cytomembrane fragments, and exosome-mimetics generated with polymers and lipids [149]. The yield of nanovesicles by a serial extrusion method was 500-fold higher than exosomes [150].

Secondly, poor drug-loading efficiency and uncontrolled drug release also limit the therapeutic application of cell- and EV-based drug delivery systems [11,38]. Loading drugs into nano vehicles before subjecting them to internalizing by carrier phagocytes could improve the loading efficiency while reducing the cytotoxicity on the carrier cells [22,42]. Fusing tumor-derived extracellular vesicles with phospholipids to form hybrid nanovesicles could improve the drug-loading capacity and stability, which represents a feasible strategy [151].

Thirdly, besides improving drug-loading efficiency, minimizing the impact of the drug-loading processes and the loaded drugs on carrier cells is another noteworthy issue. The complex fabrication processes may affect the in vivo kinetic characteristics of the carrier cells, and therefore hamper the delivery efficiency and the therapeutic effect [10]. In addition, microenvironment-responsive prodrugs with low toxicity can be developed and loaded into the carrier cells. After reaching the targeted disease lesions, the prodrugs can be released and converted to active metabolites in response to specific pathological microenvironments (i.e., pH, redox, or enzyme).

Fourthly, short circulation time and insufficient targeting capability are also typical drawbacks of natural EV-based drug delivery systems. Bioengineering strategies employed to improve these adverse properties include genetic engineering, click chemistry, cloaking, and bio-conjugation [11,64]. Surface modification of polyethylene glycol (PEG) or CD47 (a potent “do-not-eat-me” signal) was employed to inhibit the uptake of the mononuclear phagocyte system, thus prolonging the circulation time and increasing the accumulation in lesioned tissues [152,153].

It should be pointed out that there could be potential side effects of cell- and EV-based drug delivery systems. Given that tumor cell-derived EVs (TDEVs) participate in cancer progression and metastasis [154], the application of TDEVs for drug delivery may promote cancer development. Reassembling TDEVs to discard their oncogenic content could reduce their tumor-promoting side effects while retaining the intrinsic homing ability [155]. There are also safety concerns about some carrier cells, such as the differentiation of stem cells and macrophages after reinfusion in patients, which may result in unpredictable consequences [10]. Since tumor-infiltrating macrophages can differentiate into protumoral tumor-associated macrophages rapidly at both primary and metastatic sites, thus promoting tumor progression [156]. The safety issue needs to be investigated before clinical application.

Overall, the application of cells or cell derivatives as Trojan delivery systems for the diagnoses and treatments of brain diseases is a new field. Multidisciplinary efforts should be made to achieve large-scale production, develop standardized drug-loading and final product quality control methods, optimize the pharmacokinetics, and demonstrate safety properties to promote clinical translation.

## Figures and Tables

**Figure 1 pharmaceutics-15-01257-f001:**
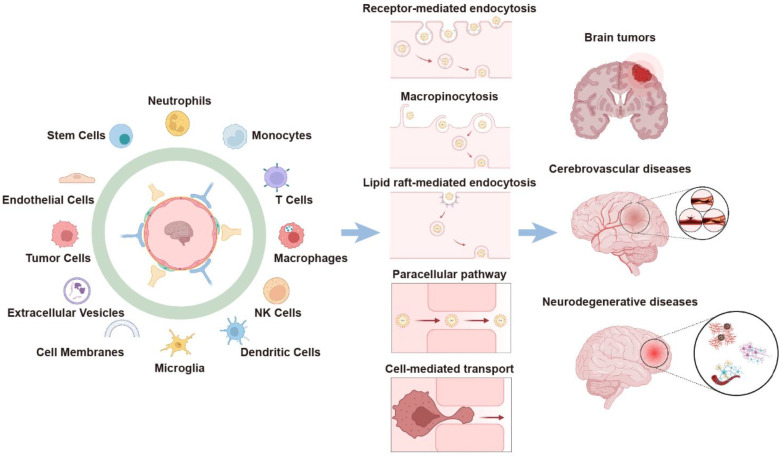
Schematic illustration of various “Trojan horses” delivery systems for BBB-penetrating delivery for brain diseases. Elements of this figure were created from BioRender.com with permission.

**Figure 2 pharmaceutics-15-01257-f002:**
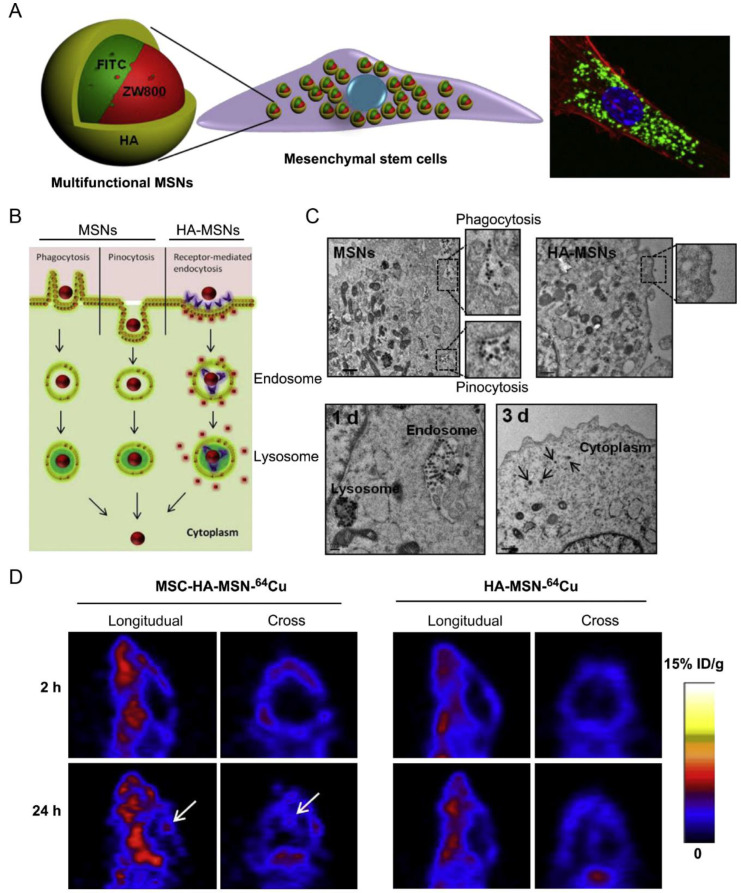
(**A**) Schematic of the structure of MSNs. (**B**) Schematic of the uptake mechanism. (**C**) TEM of the uptake mechanism. (**D**) PET imaging of the MSN-mediated tumor targeting. HA, hyaluronic acid-based polymer; MSNs, mesoporous silica nanoparticles; TEM, transmission electron microscopy; PET, positron emission tomography. Reprinted with permission from Ref. [28], Copyright^©^ 2012 Published by Elsevier Ltd.

**Figure 3 pharmaceutics-15-01257-f003:**
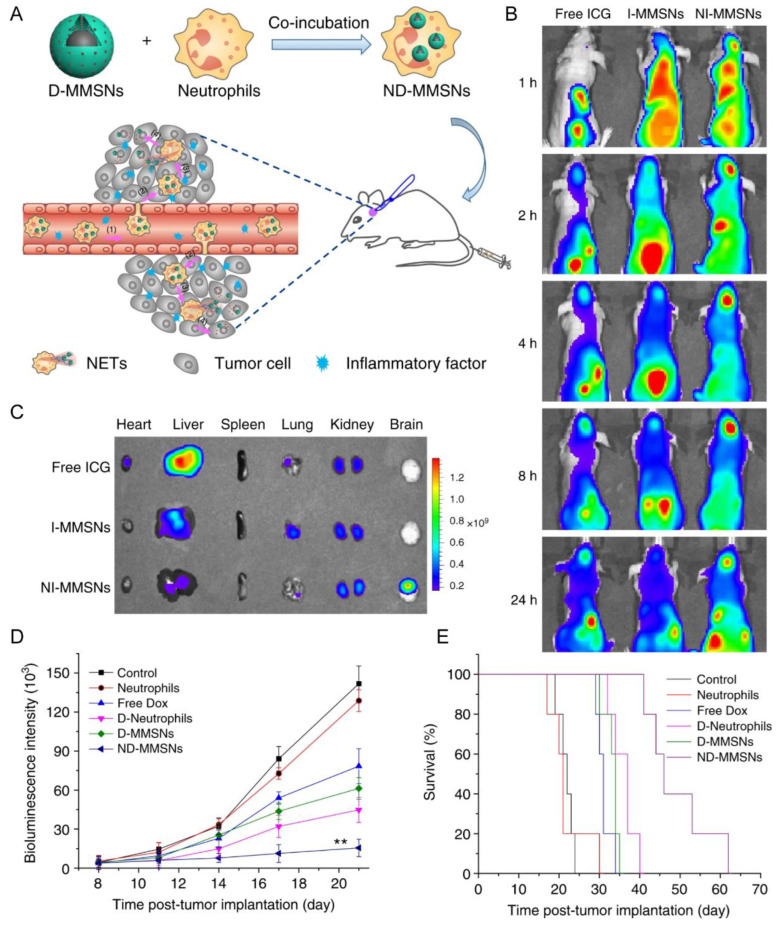
(**A**) Schematic synthesis of the ND-MMSNs for targeting the inflamed glioma sites. (**B**) In vivo fluorescence images of U87 glioma-bearing mice after intravenous administration of NI-MMSNs (I, near-infrared dye indocyanine green (ICG)). (**C**) Ex vivo fluorescent images of major organs. (**D**) In vivo therapeutic efficacy and (**E**) survival curves of ND-MMSNs in mice bearing U87 glioma. ** *p* < 0.01, as assessed by Student’s two-sided *t*-test compared to the control group. MMSNs, magnetic mesoporous silica nanoparticles; D-MMSNs, doxorubicin-loaded MMSNs; ND-MMSNs, neutrophils internalizing D-MMSNs; NETs, neutrophil extracellular traps; I-MMSNs, MMSNs fluorescently labeled with near-infrared dye ICG; NI-MMSNs, I-MMSN-internalized neutrophils; Dox, doxorubicin. Reprinted with permission from Ref. [33], Copyright^©^ 2018, Springer Nature.

**Figure 4 pharmaceutics-15-01257-f004:**
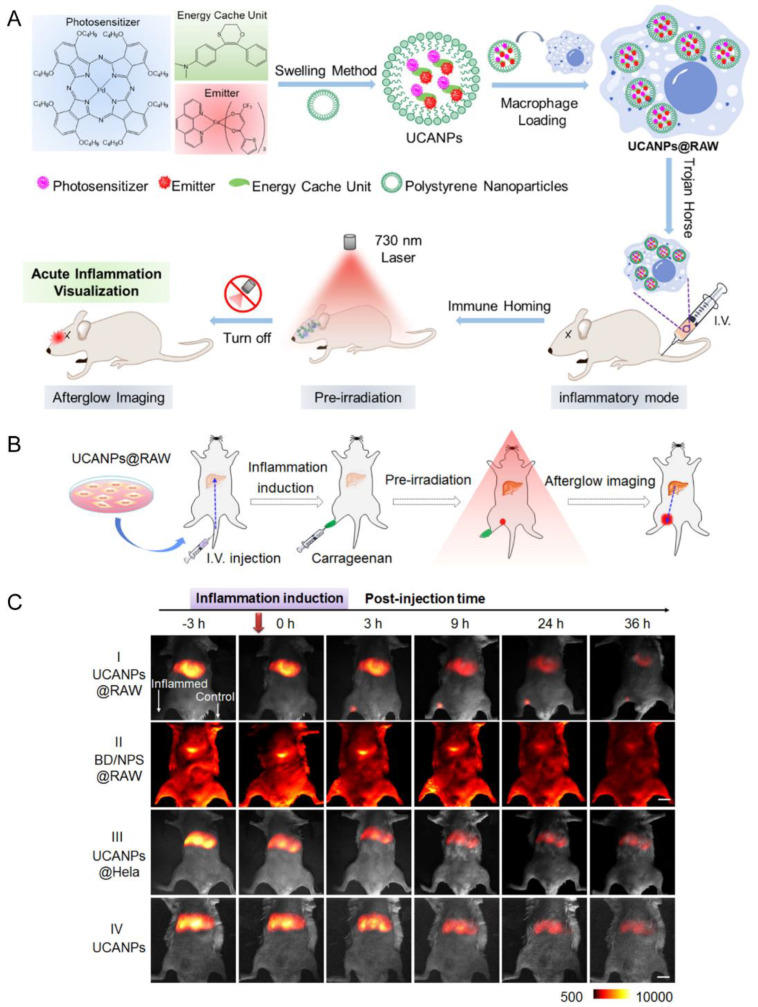
(**A**) Schematic preparation of the UCANPs@RAW for application in visualizing acute inflammation. (**B**) The treatment scheme of UCANPs@RAW. (**C**) In vivo afterglow and fluorescence imaging of UCANPs@RAW in the acute inflammation model. UCANPs, a photochemical afterglow probe; UCANPs@RAW, macrophage-camouflaged afterglow nanocomplex; BD/NPs@RAW, macrophage-camouflaged Bodipy-based nanocomplex; UCANPs@Hela, Hela cell-camouflaged nanocomplex. Reprinted with permission from Ref. [48], Copyright^©^ 2022, American Chemical Society.

**Figure 5 pharmaceutics-15-01257-f005:**
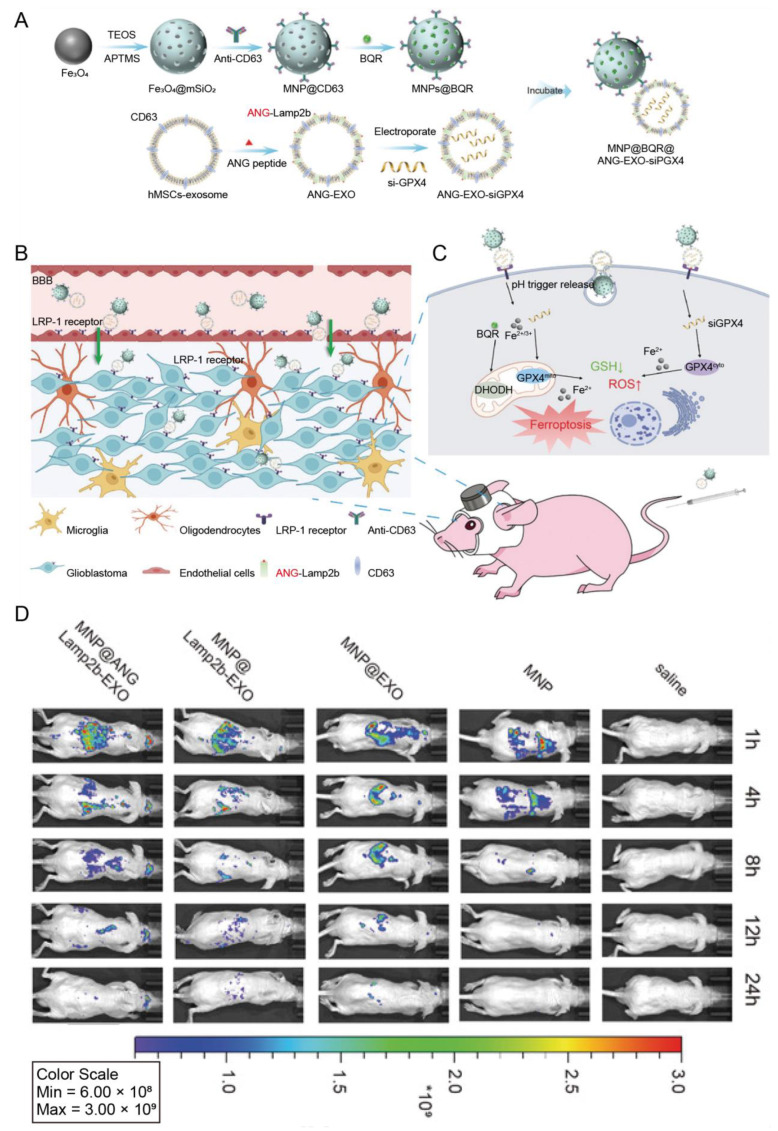
Mesenchymal stem cell-derived exosomes combined with magnetic nanoparticles for targeting glioblastoma therapy. (**A**) Schematic illustration of the design and synthesis of MNP@BQR@ANG-EXO-siGPX4. (**B**) Schematic of the magnetic mouse helmet and the mechanisms of ANG peptide-mediated NPs penetrating the BBB and accumulation in tumors. (**C**) Mechanisms that induce ferroptosis in glioblastoma (GBM) cells. (**D**) In vivo distribution of saline, MNPs, MNP@EXO, MNP@Lamp2b-EXO, and MNP@ANG-EXO in orthotopic diffuse intrinsic pontine glioma (DIPG)-bearing mice at 24 h postinjection. TEOS, silicon tetraacetate; APTMS, (3-aminopropyl) trimethoxysilane; Fe_3_O_4_@mSiO_2_, Fe_3_O_4_ nanoparticles@mesoporous silica; MNP, magnetic nanoparticles; BQR, brequinar; ANG, angiopep-2 peptide (TFFYGGSRGKRNNFKTEEYC); EXO, exosome; GPX4, glutathione peroxidase 4; si-GPX4, small interfering RNA (siRNA) of GPX4; LRP-1, low-density lipoprotein receptor protein 1; DHODH, dihydroorotate dehydrogenase; GSH, glutathione; ROS, reactive oxygen species. Reprinted with permission from Ref. [75], ^©^2022 The Authors. Advanced Science published by Wiley-VCH GmbH.

**Figure 6 pharmaceutics-15-01257-f006:**
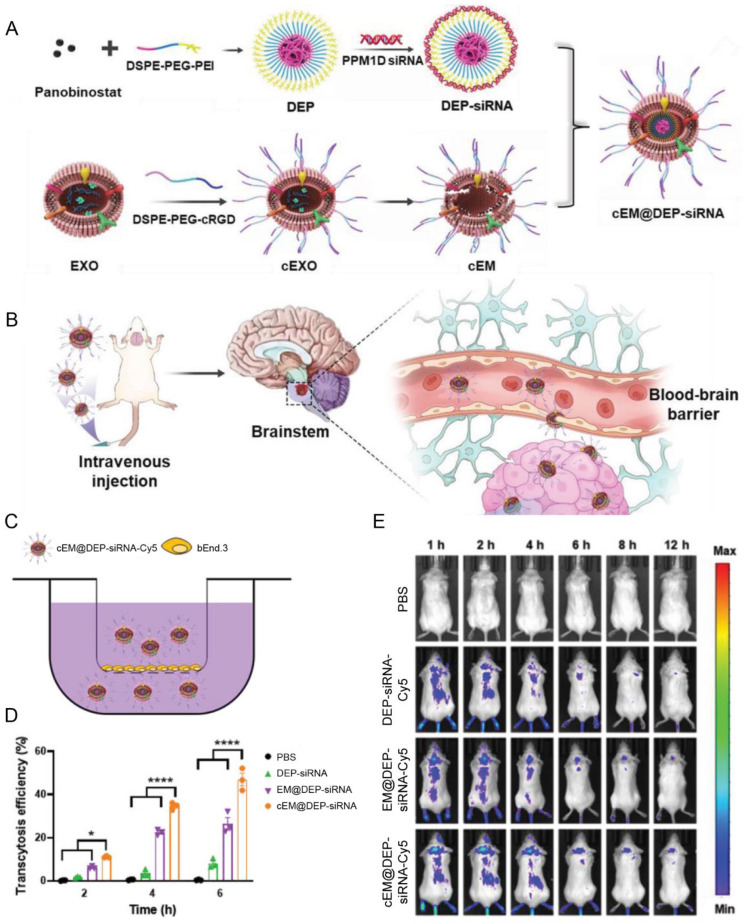
Macrophage-derived EVs for glioma therapy. (**A**) Schematic illustration of the process of exosome-based drug delivery system. (**B**) The cEM@DEP-siRNA penetrated the BBB and achieved tumor-targeting effects via tail injection. (**C**) Illustration of the in vitro BBB model. (**D**) Transcytosis efficiency of DEP-siRNA, EM@DEP-siRNA, and cEM@DEP-siRNA in the in vitro BBB model. (**E**) In vivo distribution of PBS, DEP-siRNA-Cy5, EM@DEPsiRNA-Cy5, and cEM@DEP-siRNA-Cy5 in orthotopic DIPG-bearing mice at 12 h postinjection. * *p* < 0.05, **** *p* < 0.0001, as assessed by one-way ANOVA. DEP, nanomicelles sealing panobinostat; PPM1D, protein phosphatase, magnesium-dependent 1, delta; EXO, exosome; cRGD, cRGD, cyclo (Arg-Gly-Asp-D-Tyr-Lys); cEXO, cRGD-functionalized EXO; EM, exosomal membranes from EXO; cEM, exosomal membranes from cEXO. Reprinted with permission from Ref. [92], ^©^2022 The Authors. Advanced Science published by Wiley-VCH GmbH.

**Figure 7 pharmaceutics-15-01257-f007:**
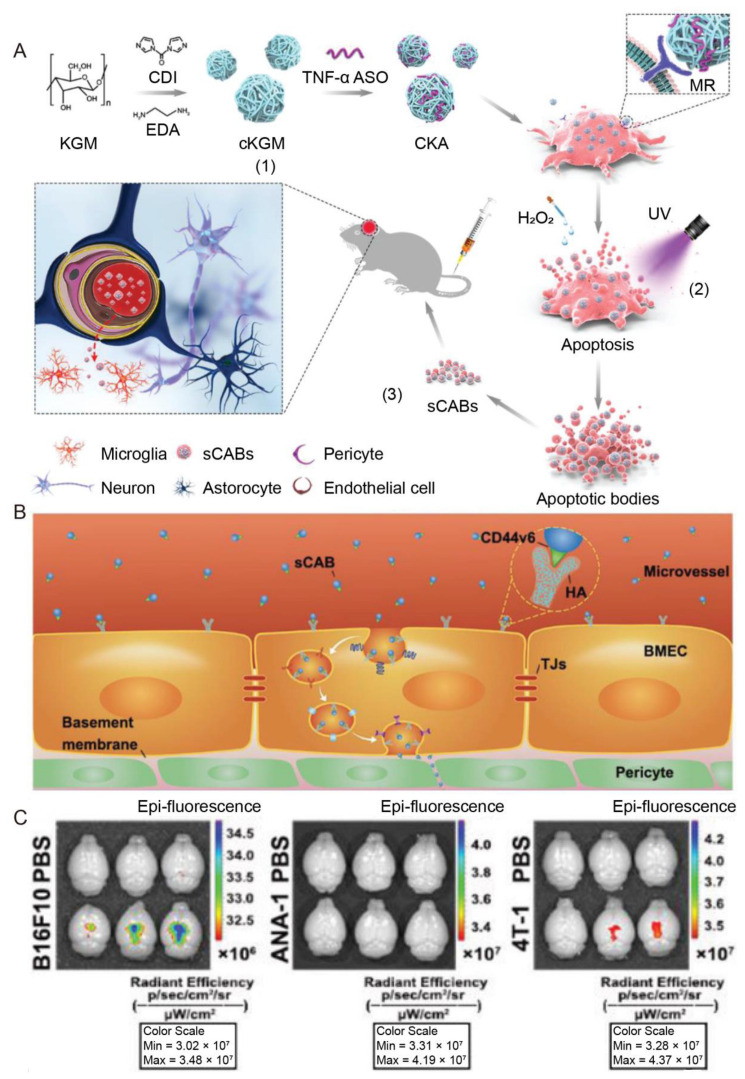
Tumor-derived EVs delivered therapeutic cargo for Parkinson’s disease treatment. (**A**) Schematic diagram of producing small apoptotic bodies (sCABs) loaded with ASO for brain penetration. (**B**) Schematic diagram of brain microvessel endothelial cells transcytosis of sCABs. (**C**) The comparison of the BBB penetrating efficiency of EVs derived from three different tumor cells. KGM, konjac glucomannan; cKGM, cationic konjac glucomannan; CDI, N,N′-carbonyldiimidazole; EDA, ethylenediamine; ASO, antisense oligonucleotide; TNF-α ASO, anti-TNF-α antisense oligonucleotide; CKA, cKGM/ASO complex; MR, mannose receptor; sABs, small apoptotic bodies; sCABs, ASO-loaded sABs; HA, hemagglutinin; TJ, tight junction; BMEC, brain microvascular endothelial cell. Reprinted with permission from Ref. [95], ^©^2021 The Authors. Advanced Science published by Wiley-VCH GmbH.

**Figure 8 pharmaceutics-15-01257-f008:**
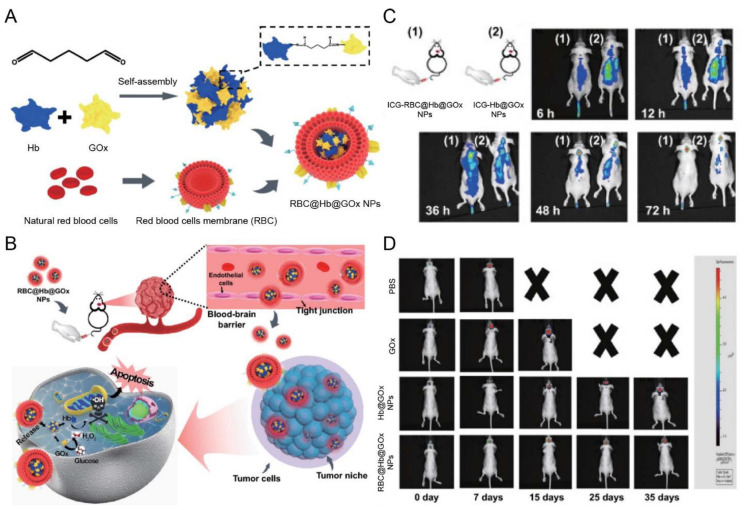
(**A**) Schematic synthesis routes of the RBC@Hb@GOx NPs. (**B**) Scheme of TME-activated ROS production by RBC@Hb@GOx NPs for GBM treatment. (**C**) In vivo biodistribution of ICG-RBC@Hb@GOx NPs and ICG-Hb@GOx NPs in mice bearing orthotopic U87MG tumor. (**D**) In vivo therapeutic efficacy of RBC@Hb@GOx NPs. Hb, hemoglobin; GOx, glucose oxidase; TME, tumor microenvironment; ROS, reactive oxygen species; GBM, glioblastoma. Reprinted with permission from Ref. [122], Copyright^©^ 2020, Springer Nature.

**Figure 9 pharmaceutics-15-01257-f009:**
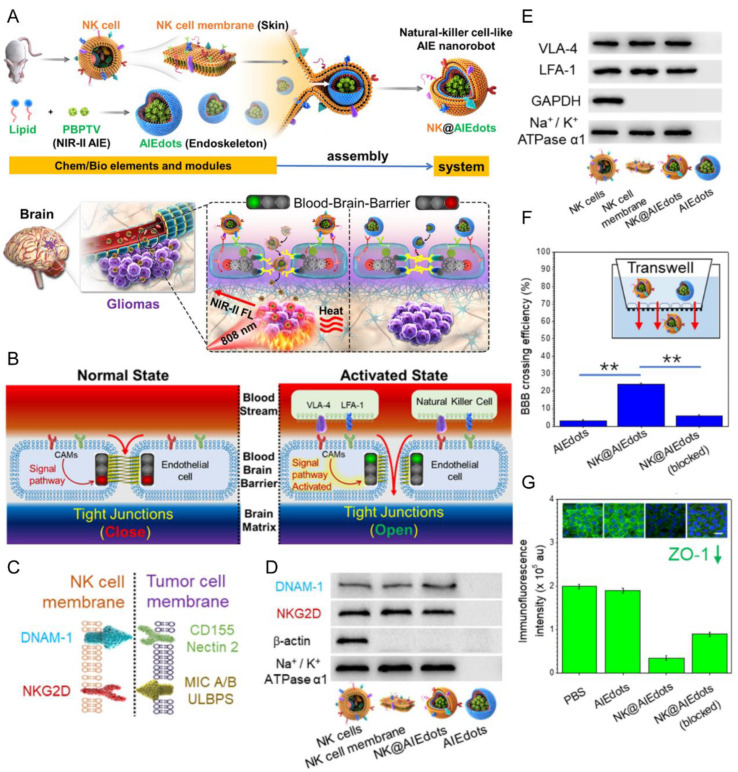
(**A**) Schematic illustration of the synthesis and enhanced BBB penetration of NK cell membrane-coated AIEdots. (**B**) Scheme and principal of NK cell-mediated BBB penetration. (**C**) Scheme of NK cell membrane-mediated brain tumor targeting. (**D**)The presence of DNAM-1 and NKG2D on NK@AIEdots. (**E**) The presence of VLA-4 and LFA-1 on NK@AIEdots. (**F**) BBB penetrating efficiency of NK@AIEdots. (**G**) Immunofluorescence staining of TJ-associated ZO-1 after NK@AIEdots treatment. ** *p* < 0.01. NK cell, natural killer cell; AIE, aggregation-induced emission; PBPTV, a NIR-II AIE-active conjugated polymer; CAMs, cell adhesion molecules; DNAM-1, DNAX accessory molecule (CD226); NKG2D, natural killer group 2D; LFA-1, lymphocyte function-associated antigen 1; VLA-4, very late antigen-4; ZO-1, Zonula occludens-1. Reprinted with permission from Ref. [126], Copyright^©^ 2020, American Chemical Society.

**Figure 10 pharmaceutics-15-01257-f010:**
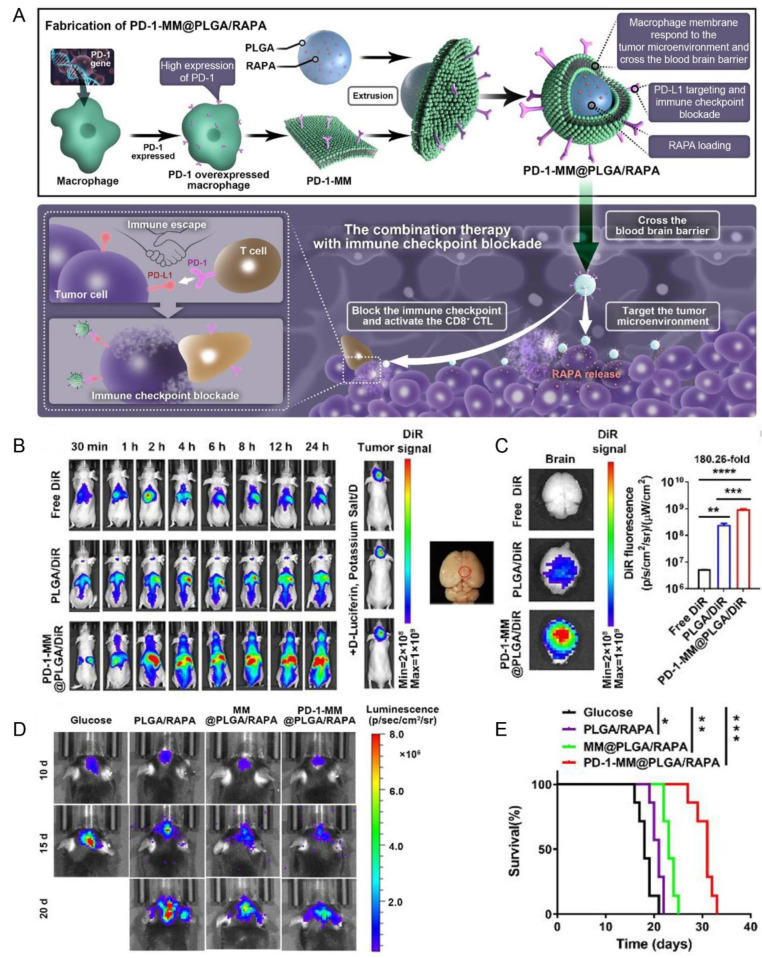
(**A**) Schematic illustration of the synthesis and proposed mechanism of PD-1-engineered macrophage membrane-coated nanoparticles. (**B**) In vivo bioluminescent fluorescence images of PD-1-MM@PLGA/DiR NPs in mice bearing orthotopic C6-luc glioblastoma. (**C**) Ex vivo the accumulation of PD-1-MM@PLGA/DiR NPs in the brain. (**D**) Whole-body imaging of the bioluminescent fluorescence and (**E**) survival curve of C6 glioma mice after PD-1-MM@PLGA/RAPA NP treatment. * *p* < 0.05; ** *p* < 0.01; *** *p* < 0.001; **** *p* < 0.0001. PD-1, programmed cell death-1; PD-L1, programmed cell death-ligand 1; MM, macrophage membrane; PLGA, poly(lactic-co-glycolic acid); RAPA, rapamycin; CTL, cytotoxic T-lymphocyte. Reprinted with permission from Ref. [132], Copyright^©^ 2022, American Chemical Society.

**Figure 11 pharmaceutics-15-01257-f011:**
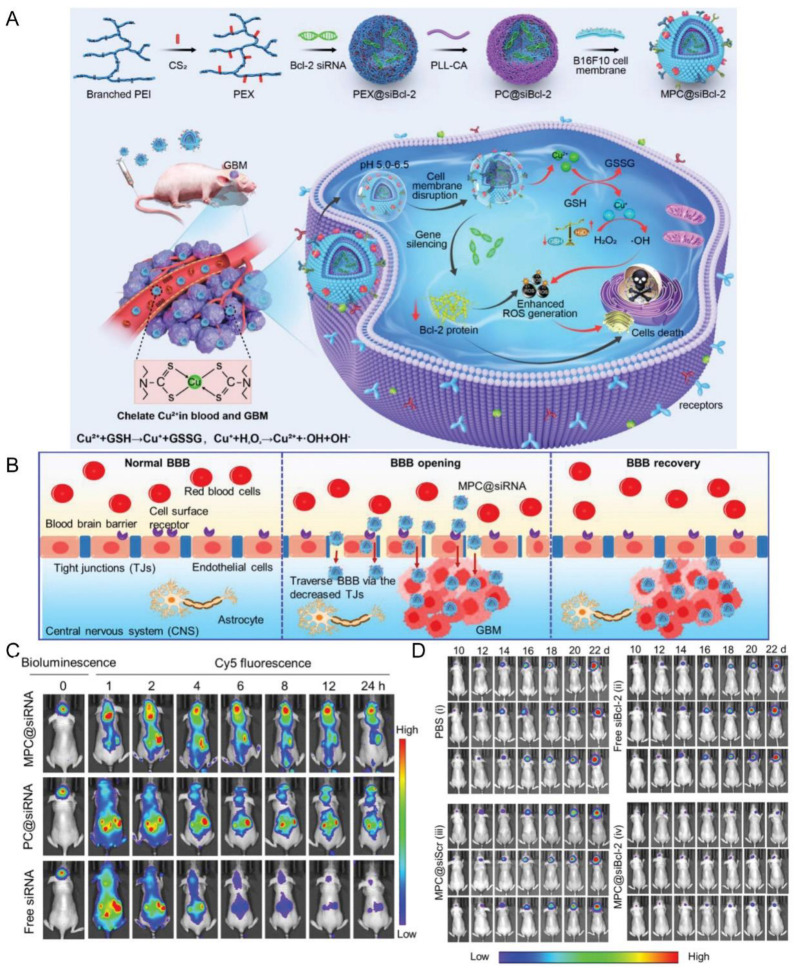
(**A**) Schematic illustration of the PC@siRNA nanocomplexs coated with B16F10 cell membranes for the combination of chemodynamic therapy (CDT) and RNAi in orthotopic glioblastoma. (**B**) Schematic diagram of nanocomplex-mediated BBB penetration. (**C**) In vivo imaging of Cy5-labeled MPC@siRNA in orthotopic U87MG-Luc GBM tumor model. (**D**) In vivo anti-GBM performance of MPC@siBcl-2. CS_2_, carbon disulfide; PEX, polyethyleneimine xanthate; Bcl-2, B-cell lymphoma 2; PLL-CA, citraconic anhydride grafted poly-lysine; GBM, glioblastoma; GSH, glutathione; GSSG, glutathione disulfide; ROS, reactive oxygen species. Reprinted with permission from Ref. [137], ^©^2022 The Authors. Advanced Functional Materials published by Wiley-VCH GmbH.

## Data Availability

Not applicable.

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
