# Peer review of "Living Cells and Cell-Derived Vesicles: A Trojan Horse Technique for Brain Delivery"

_pharmaceutics, 2023, doi:10.3390/pharmaceutics15041257_

Round 1
Reviewer 1 Report
The manuscript by Ante Ou et al. is a comprehensive review of the immense material published in the field of concern over the past five years. The review is well structured, and the conclusion critically evaluates the state of the art and future prospects for the clinical implementation of the strategy. I have only minor comments related to the Journal's format requirements, and a few recommendations.
- References should be given in square brackets.
- Fig. 1. Cell-mediated transcytosis is presented incorrectly. Please correct. Also, all text designations should be made larger.
- Fig. 2. What is HA? Apparently, this is hyaluronic acid polymer. Please explain different processes of endocytosis as in the citing reference. Fig. 2D: “MSN-HA-MSN..” should be substituted for “MSC-HA-MSN…” as in the original paper.
The resolution in this picture should be improved.
- Fig. 4C. Abbreviations should be deciphered in the figure caption, as in the original article.
- Tables 1, 2, 3. In the column “Administration method”, “Intravenous injections” should be better substituted for “IV”.
- Note to the Table 1. In “(Arg-Gly-Asp-D-Tyr-Lys)”, it is better to make the letter D a reduced size.
- Page 16, line 25: “method has facile..” should be substituted for “method is facile..”
- Fig. 7. The resolution of the drawing should be improved. There is no deciphering of abbreviations.
- Page 21, lines 226-227: the verb is missing from the sentence.
- Table 2. Ref. 106 does not match. Please check the correspondence of all references in the tables.
- Page 30. Title 4.2: Immune cells (plural).
- Page 39. Please fix formatting.
- Page 46. References 136, 137, 139: please fix formatting.
Reviewer 2 Report
Please re-new bibliography
Reviewer 3 Report
Well written and interesting review. The text is clear and well organized. Detailed notes below.
1. Page 1, Line 4: It is not known who wrote this article. The name of the senior author appears to have been lost. The corresponding author has not been indicated.
2 2. The descriptions in Figure 1 are illegible due to the small font size. The same remark applies to Figures 5-8, 10. I don't know how it will be possible to improve the quality of these Figures. They are reprints.
3. Page 5, Line 132: unnecessary space in "cell-based".
4 4. Page 16, Lines 3-4: Too many subpopulations of EVs have been listed in my opinion. Usually three are distinguished: exosomes, ectosomes also called microvesicles, and apoptotic bodies. I don't know what the term "membrane vesicles" means in the context of a subpopulation of EVs.
5 5. Page 19, Lines 173-174: Apoptotic bodies are released by cells undergoing apoptosis. Their origin is not related to the budding of the cell membrane. Such a mechanism applies to ectosomes.
6 6. Page 21, Line 236: Please convert Evs to EVs.
7 7. Page 31, Line 71: Since integrins are heterodimeric transmembrane cell adhesion molecules made up of alpha and beta subunits, it should be alpha 4 beta 1 integrin in brackets, not the listed integrin subunits, i.e. alpha 4 and beta 1.
8 8. Table 3: Other column widths - one or two last letters of words go to the line below. I propose to orient the table horizontally, not vertically. List of abbreviations used in Table 3 written in very large font.
9 9. Page 39, Line 196: I have the impression that this is where the Conclusions and further perspectives section begins. If I'm right, please put the appropriate headline.
